# Natural variation reveals that intracellular distribution of ELF3 protein is associated with function in the circadian clock

Muhammad Usman Anwer[1,2], Eleni Boikoglou[1,3†], Eva Herrero[1‡], Marc Hallstein[1], Amanda Melaragno Davis[1,2], Geo Velikkakam James[1], Ferenc Nagy[3], Seth Jon Davis[1,2]*

[1]Department of Plant Developmental Biology, Max Planck Institute for Plant Breeding Research, Cologne, Germany; [2]Department of Biology, University of York, York, United Kingdom; [3]Institute of Plant Biology, Biological Research Centre of the Hungarian Academy of Sciences, Szeged, Hungary

**Abstract** Natural selection of variants within the *Arabidopsis thaliana* circadian clock can be attributed to adaptation to varying environments. To define a basis for such variation, we examined clock speed in a reporter-modified Bay-0 x Shakdara recombinant inbred line and localized heritable variation. Extensive variation led us to identify *EARLY FLOWERING3 (ELF3)* as a major quantitative trait locus (QTL). The causal nucleotide polymorphism caused a short-period phenotype under light and severely dampened rhythm generation in darkness, and entrainment alterations resulted. We found that ELF3-Sha protein failed to properly localize to the nucleus, and its ability to accumulate in darkness was compromised. Evidence was provided that the *ELF3-Sha* allele originated in Central Asia. Collectively, we showed that ELF3 protein plays a vital role in defining its light-repressor action in the circadian clock and that its functional abilities are largely dependent on its cellular localization.

*For correspondence: seth. davis@york.ac.uk

Present address: †Cold Spring Harbour Laboratory, Laurel Hollow, Germany; ‡MRC National Institute for Medical Research, London, United Kingdom

Competing interests: The authors declare that no competing interests exist.

## Introduction

The rotation of the earth leads to environmental changes in ambient light and temperature that define a 24-hr cycle. During these diurnal cycles, light is correlated with warmth, while darkness is correlated with coolness. Thus, to ensure fitness in response to such regular environmental oscillations, many organisms have evolved an internal timing mechanism to predict these cyclic environmental cues, and this is termed the circadian clock. This internal clock synchronizes major developmental and physiological processes. For example, a considerable proportion of metabolism is under clock control (*Graf et al., 2010*; *Haydon et al., 2011*; *Sanchez et al., 2011*; *Sanchez-Villarreal et al., 2013*), and seasonal developmental programs require a circadian oscillator (*Thines and Harmon, 2011*). The coordination of molecular and physiological processes with the external environment thus provides an adaptive advantage under diverse climatic conditions (*Dodd et al., 2005*).

Circadian rhythms are produced by molecular oscillators, which are comprised of interlocking feedback loops having several components (*Bujdoso and Davis, 2013*). In Arabidopsis, the central loop consists of two closely related single-myb transcription factors, CIRCADIAN CLOCK ASSOCIATED 1 (CCA1) and LATE ELONGATED HYPOCOTYL (LHY), as well as the pseudo response regulator (PRR) TIMING OF CAB EXPRESSION 1 (TOC1). In the morning, CCA1 and LHY repress *TOC1* by directly binding to its promoter, resulting in the evening accumulation of TOC1, which, in turn, represses *CCA1* and *LHY* expression (*Gendron et al., 2012*; *Huang et al., 2012*). The core oscillator is further tuned by an evening- and morning-phased loop. The evening loop is proposed to include GIGANTEA (GI) and TOC1, where GI activates the expression of *TOC1*, and TOC1 represses *GI* (*Locke et al., 2006*;

**eLife digest** Life on Earth tends to follow a daily rhythm: some animals are awake during the day and asleep at night, whilst others are more active at night, or during the twilight around dawn and dusk. For many living things, these cycles of activity are driven by an internal body clock that helps the organism to adapt to the daily cycle of light and dark—and similar internal clocks also exist in plants.

These internal clocks define daily—or circadian—cycles whereby multiple genes are switched 'on' or 'off' at different time points in every 24-hr period. And, because light and ambient temperatures also vary with time of the day, many organisms use these external signals as cues to reset their own internal clocks. Moreover, the hours of daylight and temperature vary around the world, and also with the seasons, so plants and animals must be able to change how these external signals influence their internal clocks so that they stay in tune with the day/night cycle. However, it is not clear how they do this.

To explore this question, Anwer et al. grew plants that were from a cross between two types of the model plant *Arabidopsis thaliana* from different environments: one from Germany, and the other from Tajikistan in Central Asia. These offspring were also genetically engineered so that an enzyme that could give off light was produced under the control of the internal clock. Anwer et al. found that the plants continued to glow and fade with an almost daily rhythm even after external cues, such as changes in temperature or light, had been removed.

Different offspring plants consistently glowed and faded with different rhythms such that some had, for example, a 21-hr day and others a 28-hr day. These differences were caused by many genes that differed from the original German and Tajikistan parent plants, and Anwer et al. 'mapped' one of these genetic differences to a single gene. Offspring that inherited a version of a gene called *ELF3* from the Tajikistan parent had internal clocks that ran faster when the plant was under the light. These plants also gradually stopped glowing as brightly as the German parent when they were kept in the dark, suggesting that their internal clocks were 'ticking more softly'. It was already known that the *ELF3* gene affected the circadian clock in plants, and Anwer et al. thus concluded that the plants with Tajikistan version of this gene, called *ELF3-Sha*, were also less able to reset their internal clocks to synchronize in response to external cues.

Anwer et al. also showed that the normal ELF3 protein is more likely to be found in the nucleus of a plant cell than the ELF3-Sha version, which might suggest that this protein is involved in switching genes off. Further research is now needed to uncover exactly how the ELF3 protein does this to keep the plant's internal clock 'ticking' correctly.

---

*Zeilinger et al., 2006*; *Huang et al., 2012*). Two members of the pseudo response regulator gene family, *PRR7* and *PRR9*, repress *CCA1* and *LHY* during the day, and this establishes the morning loop (*Locke et al., 2006*; *Zeilinger et al., 2006*). Recently, an evening complex comprising EARLY FLOWERING 3 (ELF3), EARLY FLOWERING 4 (ELF4), and LUX ARRYTHMO (LUX) has been found to repress the morning loop by specifically binding to the *PRR9* promoter to mediate transcription repression (*Nozue et al., 2007*; *Kolmos et al., 2009*; *Dixon et al., 2011*; *Helfer et al., 2011*; *Nusinow et al., 2011*; *Chow et al., 2012*; *Herrero and Davis, 2012*; *Herrero et al., 2012*).

Ambient light and temperature are two important environmental factors, termed *zeitgebers*, which reset the clock in a process referred to as entrainment (*McClung and Davis, 2010*; *McClung, 2011*). PRR7 and PRR9 have been reported to play a role in temperature compensation, which is the resistance of period change under differing mean ambient temperatures (*Salomé and McClung, 2005*; *Salome et al., 2010*). However, diurnal temperature regulation of the circadian clock is still poorly understood (*McClung and Davis, 2010*). In contrast, light has been established as a key factor for the entrainment of the clock where continuous irradiation shortens free-running period length in a fluence-rate-dependent manner as a process termed parametric entrainment (*Aschoff, 1979*; *Covington et al., 2001*; *Johnson et al., 2003*). Most genetic components of the clock were originally discovered from a light-entrained oscillator. Several studies have shown that many observable rhythms in plants dampen in constant darkness (DD), where dampening is defined as a reduction in circadian amplitude. For example, many genes have been associated with circadian rhythms of mRNA abundance that are expressed

robustly under constant light (LL), but dampen in DD (*Watillon et al., 1993*; *Zhong et al., 1997*). Furthermore, the cyclic protein accumulation of several clock-regulated components, such as GI and ELF3 (*Liu et al., 2001*; *David et al., 2006*), was found to be reduced when shifted to darkness. Thus, the effects of various light-input signals on the Arabidopsis core oscillator remain to be elucidated.

Molecular and genetic studies support the involvement of ELF3 in light gating to the circadian clock, and ELF3 is also a required component of the core oscillator (*Covington et al., 2001*; *Thines and Harmon, 2010*; *Kolmos et al., 2011*; *Herrero et al., 2012*). The *elf3* mutant was originally isolated in a screen for lines displaying photoperiod-independent early flowering (*Zagotta et al., 1992*, *1996*). Further characterization of *elf3* revealed other severe phenotypes, including defects in clock-regulated leaf movement rhythms, rhythmic hypocotyl elongation, and arrhythmic expression of gene expression under the free running condition of LL and in DD (*Hicks et al., 1996*; *Reed et al., 2000*; *Thines and Harmon, 2010*; *Kolmos et al., 2011*; *Herrero et al., 2012*). Natural variation at *ELF3* has been detected that could be associated to an alteration in shade-avoidance responses, which included increased elongation of the hypocotyl, delay of cotyledon opening in seedlings, increased elongation of stems and petioles, and reduced developmental timing in adults, and to changes in circadian function (*Tajima et al., 2007*; *Jimenez-Gomez et al., 2010*; *Coluccio et al., 2011*; *Undurraga et al., 2012*). *ELF3* was cloned and reported to encode a nuclear protein of unknown function, and it was proposed to work as a transcription factor (*Hicks et al., 2001*; *Liu et al., 2001*). Based on the accumulation of ELF3 protein upon shifting plants to LL, and its decrease when plants were shifted to DD, it was concluded that its abundance is dependent on ambient light (*Liu et al., 2001*). To extend this hypothesis based on the reported light-dependent phenotypes of *elf3*, *McWatters et al. (2000)* proposed that ELF3 fulfills the so-called *zeitnehmer* concept, in that this factor acts to bridge light perception to the clock. This idea was further strengthened when ELF3 was found to physically interact with the phytochromeB photoreceptor (*Liu et al., 2001*). The previously noted role of ELF3 in light input to the circadian clock (*Kolmos et al., 2011*) is not altered by new findings that *ELF3* is core to the oscillator.

Many elements in the clock were isolated in screens for rhythm mutants using bioluminescence readout of the molecular oscillator. Worldwide, the species distribution of Arabidopsis ranges from the equator to extreme northern latitudes near the Arctic Circle (*Koornneef et al., 2004*). Therefore, as an alternative to induced mutants, another source of genetic variation can be found among naturally occurring populations of Arabidopsis. More specifically, environmental variations across and within local populations of Arabidopsis act as a discriminatory force on the gene pool from which only a few genetic variants of adaptive phenotypes will be selected and pass through reproduction. Arabidopsis populations based on Recombinant Inbred Lines (RILs) have been derived from parental accessions. Such populations are advantageous in mapping novel gene interactions. For example, quantitative trait locus (QTL) mapping was used to explain circadian parameters associated with natural variation in the circadian rhythmicity of leaf movement (*Swarup et al., 1999*; *Michael et al., 2003*; *Edwards et al., 2005*; *Anwer and Davis, 2013*). More recently, the rhythm of bioluminescence from modified firefly *LUCIFERASE* (*LUC*) genes coupled to the clock-controlled promoter was successfully used to map QTLs for circadian parameters (*Darrah et al., 2006*; *Boikoglou et al., 2011*). Thus, the luciferase-based system can be employed to accurately measure variation in circadian-rhythm parameters within RIL populations to detect phenotypically expressed genetic variation in circadian clock genes (*Anwer and Davis, 2013*).

In this study, we characterized a natural allele of *ELF3* to determine its effect on rhythmicity in the Arabidopsis circadian clock. Classical QTL mapping in a modified Bay-0 x Shakdara mapping population, followed by positional cloning, revealed an allele of *ELF3* (*ELF3-Sha*) that accelerates circadian oscillations in a light-dependent manner and a dampened oscillator in darkness. We determined that the periodicity phenotype of *ELF3-Sha* results from a single encoded amino-acid change A362V, which is an allelic variant that is largely confined to a latitudinal stripe in Central Asia. Furthermore, we identified that the circadian abnormalities in *ELF3-Sha* are associated to cellular localization defects of ELF3 protein that results in an inability to properly function under light and in darkness. Thus, by characterizing *ELF3-Sha*, we clarified the molecular mode-of-action of ELF3 in the circadian clock.

## Results

### Analysis of circadian periodicity detected QTLs in Bay-0 X Sha RILS

A total of 71 RILS derived from the genetic cross of Bayreuth-0 and Shakdara (BxS) were modified to harbor the *CCR2* promoter, also termed *GRP7* (*Loudet et al., 2002*), fused to luciferase (*CCR2::LUC*).

These lines were synchronized to respective photic and thermal entrainment, and free-running periodicity was assessed under identical conditions of constant light and constant temperature. After applying both entrainment protocols, extensive variation in periodicity was observed (*Figure 1A*; *Supplementary file 1*, *Supplementary file 2*); however, the two parental ecotypes displayed similar periodicity, irrespective of preceding entrainment. Compared to photic entrainment, a significant shift for shortened free-running periodicity of *CCR2* after thermal entrainment was observed for the parental lines, as well as RILs (*Figure 1A–C*; *Table 1*). The periodicity differences after respective photic vs thermal entrainment (Period Light Dark (LD)–Period TMP [TMP]) extended from minus 0.56 hr to plus 1.66 hr (*Figure 1B,C*; *Table 1*), with parental lines displaying periodicity differences of plus 1.84 and plus 2.15 hr for Sha and Bay-0, respectively (*Boikoglou et al., 2011*). The averaged periodicity differences of BxS lines were moderate (0.433, p<0.001) (*Table 1*). Thus, despite the transgressive variation observed in the RILs for *CCR2* periodicity and periodicity differences (LD-TMP), the averaged periodicity differences seen in the RILs were similar in magnitude as the periodicity differences seen in the parental lines.

*CCR2* free-running periodicity followed a normal distribution regardless of the entrainment *zeitgeber*, where lower and higher extreme periodicities were measured in thermal and photic entrainment, respectively (*Figure 1A*). The normally distributed phenotypes allowed us to test the fixed effects of entrainment (E) by the random effects of genotypes (G), and we found statistically significant G by E interactions (*Table 2*). The factor with the most significant effect in period variation was environment, with genotypes having a lesser, albeit highly significant effect (*Table 2*). Moreover, when the genotypic effect was compared between RILs and transformants, a far greater variation in periodicity could be observed in RILs (*Table 2*). Transformation-derived variation (position effects) was thus not a major component of detected variation. These findings suggest that both parents had been differentially selected in a number of loci for both *zeitgeber* inputs.

The significant effect of genotypes in periodicity prompted us to calculate trait heritabilities. These were found to be similar for photic (0.76) and thermal entrainment (0.73), respectively (*Table 3*). Our efforts then focused on mapping the genetic components of differential periodicity of the two inputs. This resulted in the identification of three large-effect QTLs for photic entrainment on chromosomes 1, 2, and 4. Two co-localized QTLs were found on chromosomes 2 and 4 (*Table 3*). The QTL on Chromosome 2 (Chr2) displayed the largest phenotypic effect and we thus pursued its identification.

## Fine-scale mapping identifies *ELF3* as a strong candidate for Chr2 QTL

To validate the identity of the Chr2 QTL, we generated heterogeneous-inbred families (HIFs), and near isogenic line (NIL). Three HIFs were made (57, 92 and 343; see 'Materials and methods' for construction details), each harboring either the Bay allele (HIF-Bay-0) or Sha allele (HIF-Sha) at the Chr2 locus. An analysis of the free-running period of these lines revealed that HIF-Sha always displayed a shorter period as compared to HIF-Bay-0. This period-shortening effect was independent of the preceding LD or TMP entrainment condition (*Figure 2A*). The complexities of the HIF genomic structure created the possibility for equally complex epistatic interactions that cannot be simply defined. Therefore, to circumvent the possibility of such interactions, a NIL with a small introgression of Sha at this Chr2 locus (NIL-S), in an otherwise homogeneous Bay genetic background, was generated. This line was measured for periodicity. Consistent with the HIFs, NIL-S displayed a 2-hr shorter period, as compared to Bay-0, which was only observed under LL. In DD, no significant period difference was detected (*Figure 2B*). However, in DD, the free-running profile of *CCR2::LUC* in NIL-S was different from the Bay parental line, so that, in NIL-S, *CCR2::LUC* rhythmicity lost robustness, whereas *CCR2::LUC* in Bay-0 displayed rhythms for 7 days (*Figure 3A*). Interestingly, under both LL and in DD, the Sha parental line displayed a *CCR2::LUC* profile similar to that of Bay-0, and different from NIL-S (*Figures 2B and 3A*).

We next employed a fine-mapping strategy to isolate the Chr2 QTL and delineated the region to a 40-Kb interval with nine annotated genes (*Figure 2C*). *ELF3* (AT2G25930) was the only gene in this region with a previously reported function in the Arabidopsis circadian clock. Therefore, we compared the sequence of *ELF3-Bay* and *ELF3-Sha* and identified two nonsynonymous changes in *ELF3-Sha*: an encoded alanine-to-valine transition at position 362 (A362V), and in the C-terminal encoded glutamine stretch, Sha had 8 more than Bay (*Figure 3C*; and as reported [*Tajima et al., 2007*; *Coluccio et al., 2011*]). Thus, *ELF3* appeared as a strong candidate for the Chr2 QTL.

We then tested whether *ELF3-Sha* is involved in clock-controlled physiological processes under normal growth conditions. This was particularly relevant as it has been reported that the *ELF3* allele in

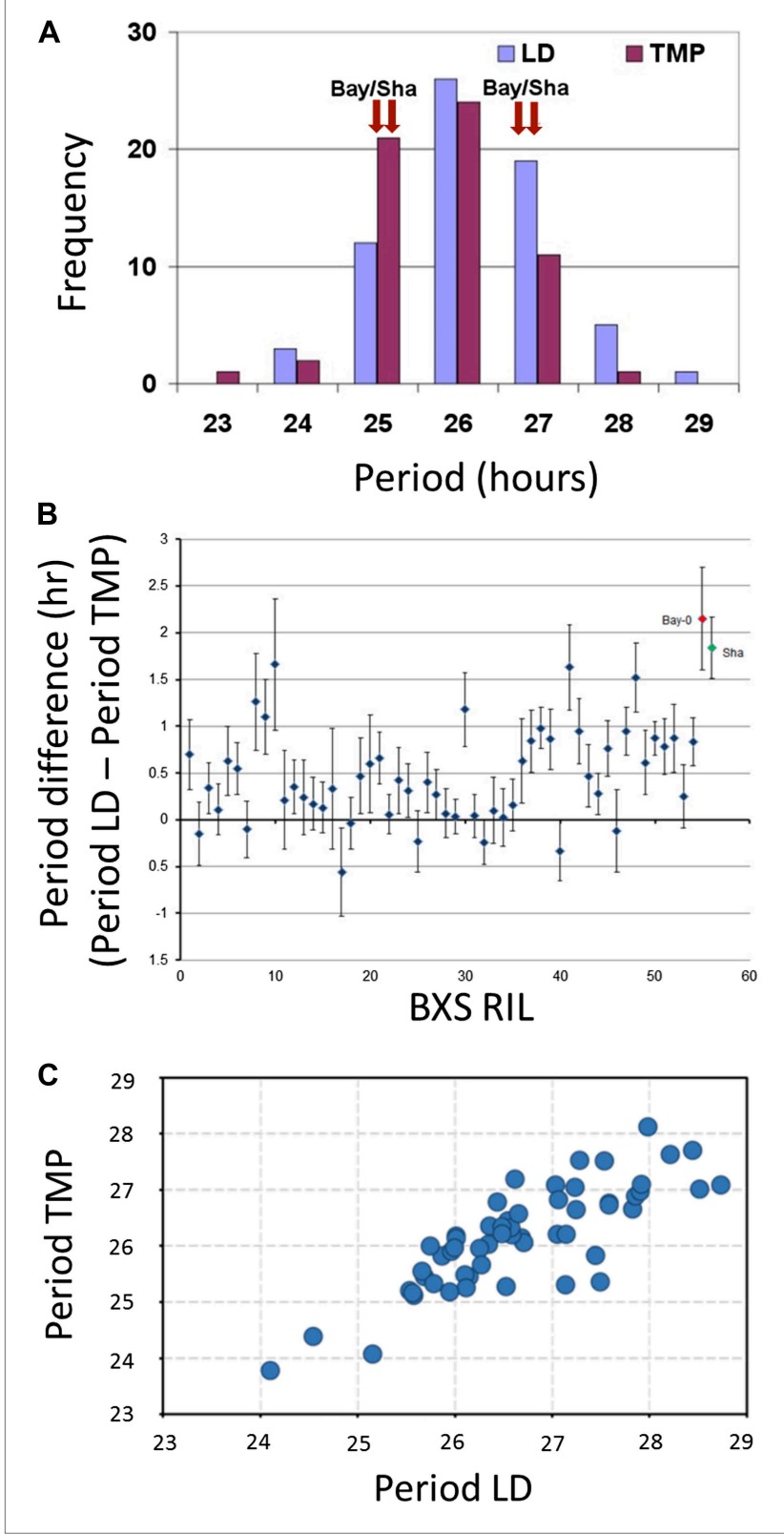

**Figure 1**. Illustrative and statistical features of *CCR2* periodicity post-photic vs post-thermal entrainment in BxS populations. (**A**) Normal frequency distribution of *CCR2* periodicity in BxS individuals. Blue-colored bars represent periodicity after photic entrainment, and magenta-colored bars represent periodicity after thermal entrainment. *Figure 1. Continued on next page*

*Figure 1. Continued*

Bay-0 and Sha denote the periodicity of *CCR2* in the parental genotypes. Note the skew of temperature-entrained plants to shorter periodicity, when compared to photic-entrained plants. (**B**) Periodicity differences of *CCR2* in BxS RIL lines. The x-axis denotes 54 RILs of which periodicity was assayed in both in LD and TMP entrainment (see *Supplementary file 1* and *Supplementary file 2*). The y-axis represents the periodicity differences of thermal minus photic-entrained lines (calculated as PeriodLD–PeriodTMP). (**C**) A scatter plot for TMP vs LD periodicities from the BxS RILs described in the tables *Supplementary file 1* and *Supplementary file 2*.

**Table 1.** Circadian periodicity analysis of the BxS RILs after photic or thermal entrainment

| | Zeitgeber | Mean* | S. E | 95% confidence interval Lower bound | Upper bound |
|---|---|---|---|---|---|
| BxS | LD | 26.702 | 0.035 | 26.633 | 26.771 |
| | TMP | 26.269 | 0.034 | 26.202 | 26.335 |

| LD-TMP | | BxS |
|---|---|---|
| Pairwise comparisons | Mean difference (hr) | 0.433 |
| | S. E | 0.047 |
| | Significance | <0.001 |
| Correlations | Correlation coefficient | 0.86 |
| | Significance | 0.001 |

LD stands for photic-*zeitgeber* and TMP for thermal-*zeitgeber*.
*denotes the modified population marginal mean for the 95% Confidence Interval.
S.E. denotes standard error.
LD-TMP denotes the pairwise comparisons such that TMP period is subtracted from LD period.

Sha affects shade-avoidance responses (*Tajima et al., 2007*; *Jimenez-Gomez et al., 2010*; *Coluccio et al., 2011*). To determine the sensitivity of *ELF3-Sha* to day length, as a basis of comparing to an *elf3* null allele, we examined the flowering time of the HIFs and NIL-S under long- and short-day growth conditions, comparing the results to the Bay-0 wild type. While we observed no large differences in flowering time under long days, HIF 89-S showed modestly advanced flowering time under short days, compared to HIF 539-B (*Figure 4A*). Next, we assessed the effect of *ELF3-Sha* on hypocotyl length by measuring the hypocotyl length of HIFs, NIL-S, Bay-0, Sha, and the null allele *elf3-1* under short days, constant red (RR), and constant blue (BB) light. Similar to the near lack of flowering-time phenotype under long days, we did not observe a substantial difference in the hypocotyl length of HIFs, NIL-S, and Bay-0 under RR or BB. Under short days, however, HIF 89-S displayed a marginally elongated hypocotyl length, compared to HIF 539-B, but this response was four times less than that seen in the null *elf3* (*Figure 4B*). Overall, a NIL that harbored *ELF3-Sha* in a Bay background did not display substantial physiological effects under normal growth conditions, indicating that *ELF3-Sha* maintains developmental activity lost in null *elf3* alleles.

## Alanine to valine substitution underlies *ELF3-Sha* circadian phenotypes

Although fine-scale mapping provided strong evidence favoring *ELF3* as the gene responsible for the short-period phenotype we observed, any one of the nine genes in the 40-kb region were candidates in the context of this study. Therefore, to generate genetic materials to test if *ELF3-Sha* was indeed the causal QTL, we cloned *ELF3*, along with its own promoter from Bay-0 and Sha, fused these with *YFP*. These constructs were then transformed to the null allele *elf3-4* in the Ws-2 background, harboring *CCR2::LUC* and *LHY::LUC* reporter genes, respectively. The free-running period analysis of T2 transgenic lines harboring *ELF3-Sha-YFP* displayed a consistent short period under LL, compared to the lines harboring *ELF3-Bay-YFP*, for both *LHY::LUC* and *CCR2::LUC* (*Figure 3B*). Based on these complementation experiments, we confirmed *ELF3* as quantitative trait gene (QTG) underlying the Chr2 periodicity QTL.

To define the causal polymorphism in *ELF3-Sha*, we investigated the role of an encoded A362V residue change compared to the difference in the number of encoded glutamines in *ELF3-Bay* and *ELF3-Sha*. In this experiment, we separately cloned the promoter and coding regions of *ELF3-Bay* (SpBc) and *ELF3-Sha* (SpSc) and then induced encoded A362V and V362A changes in *ELF3-Bay* (SpBa2v) and *ELF3-Sha* (SpSv2a) coding regions, respectively. These constructs were then recombined

**Table 2.** Statistical analysis of *CCR2* circadian periodicity after the two *zeitgeber* protocols for BxS population

| Overall model | | Univariate BxS | | |
|---|---|---|---|---|
| **Factor** | | **F** | | **P** |
| Genotype | | 7.753 | | <0.001 |
| Environment | | 36.413 | | <0.001 |
| Environment*Genotype | | 2.183 | | <0.001 |

| | | **Univariate LD** | | | **Univariate TMP** | | |
|---|---|---|---|---|---|---|---|
| | | **F** | **P** | **CV LD** | **F** | **P** | **CV TMP** |
| BxS | RIL | 10.387 | <0.001 | 18.043 | 7.382 | <0.001 | 16.498 |
| | TRANS (RIL) | 1.513 | <0.001 | | 1.915 | <0.001 | |

F = mean sum of squares\error sum of squares. P = significance value of the F-ratio.
Genotype denotes RIL, ENVIRONMENT denotes the different entrainments, TRANS denotes independent transformants of each genotype.
*denotes the testing of an interaction between two factors, whereas B(A) denotes the testing main factor A in which a factor B is nested.
CV is the coefficient of genetic variation, LD stands for photic and TMP stands for thermal entrainment.
NS denotes nonsignificance.

in all possible promoter::coding combinations (illustrated in *Figure 5A*). The free-running period of *LHY::LUC* in T2 transgenic lines harboring different coding regions, under the control of the *ELF3* promoter of the Sha accession, was analyzed under LL. We observed that lines SpSc and SpBa2v that contained encoded valine displayed a shorter period as compared to the SpBc and SpSv2a lines harboring encoded alanine, irrespective of the number of glutamines present (*Figure 5B*). We further confirmed these results in the transgenic lines expressing *ELF3-Bay* and *ELF3-A362V* under the Bay-0 *ELF3* promoter. The free-running profile of *LHY::LUC* was analyzed under LL and in DD. Both *ELF3-Bay* and *ELF3-A362V* displayed a robust rhythm of *LHY::LUC* expression under LL, albeit with lower amplitude observed in *ELF3-A362V* (*Figure 5C*). However, in DD, the expression profile of *LHY::LUC* in *ELF3-Bay* and *ELF3-A362V* was distinctly different. *ELF3-A362V* failed to maintain robust rhythms after 4 days in darkness and displayed an acute dampening of *LHY::LUC* expression, whereas *ELF3-Bay*

**Table 3.** Localization of the main QTLs for the BxS population after photic vs thermal *zeitgeber* protocols

| Zeitgeber | H² | Chromosome | Position (cM) | LOD score [*] | % expl variance | F | p value | 2a (h) |
|---|---|---|---|---|---|---|---|---|
| LD | 0.76 | I | 63.7 | 3.21 | 14.9 | 7.704 | 0.007 | −0.614 |
| | | II | 34.5 | 5.72 | 27.3 | 20.784 | <0.001 | 1.003 |
| | | IV | 69.9 | 3.89 | 17.1 | 14.003 | <0.001 | 0.833 |
| TMP | 0.73 | II | 34.5 | 4.56 | 25.5 | 25.033 | <0.001 | 0.988 |
| | | IV | 69.9 | 3.26 | 16.2 | 6.501 | 0.014 | 0.508 |

H² denotes broad sense heritability.
(cM) denotes centiMorgan.
[*] LOD- score threshold was determined at 2.4.
% expl variance is the percent of explained variance.
F = mean sum of squares\error sum of squares.
P denotes the significance value of the F-ratio.
2a denotes the additive effect of Bay allele when the effect of the Sha allele on period is subtracted.
(h) denotes the effect in hours.
− denotes that the Sha allele displayed longer period than that of Bay allele.

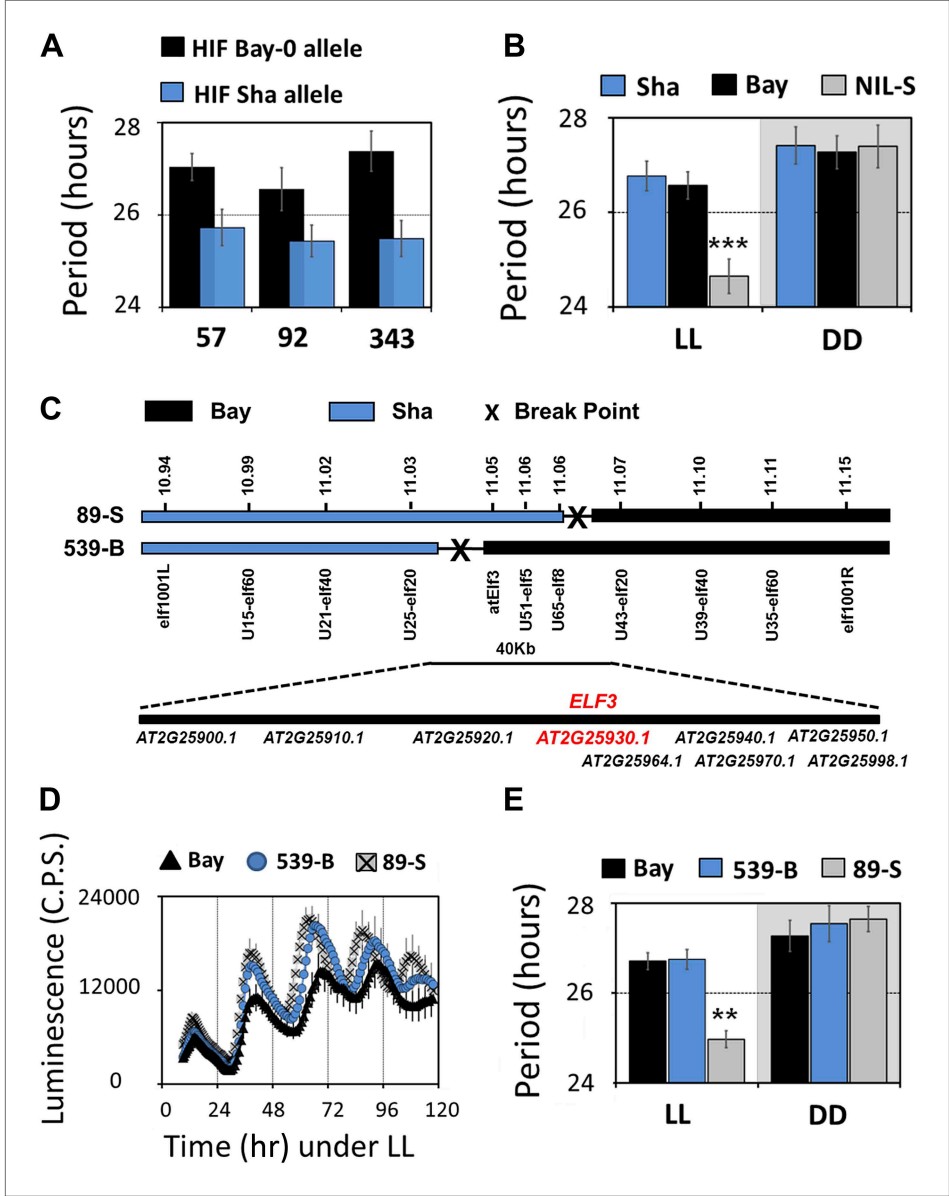

**Figure 2**. Map-based cloning identifies *ELF3* as a candidate for chr2 QTL. Period estimates of *CCR2::LUC* expression in (**A**) three independent HIFs (57, 92, and 343) harboring either the Bay-0 or the Sha allele at QTL confidence interval (**B**) parental accessions and NIL-S with introgression of Sha at QTL confidence interval in otherwise homogeneous Bay-0 background, under LL and in DD. (**C**) Schematic diagram showing the fine mapping strategy of chr2 locus. Black and gray bars represent Bay-0 and Sha genotypes, respectively. The names below the bars represent the molecular markers used for genotyping, and the numbers above correspond to their physical position on the genome. The crosses represent the position of the recombination event. Two recombinants, 89-S and 539-B, were found to have a recombination event surrounding a 40-Kb region, where nine annotated genes are located, as indicated below the solid bar. (**D**) Free-running profile of *CCR2::LUC* expression in recombinants 89-S, 539-B and Bay-0 under continuous red and blue light (LL). (**E**) Period estimates of rhythm shown in (**D**). All error bars indicate SEM, where n ≥ 24. Mean values that are significantly different from Bay-0 wild type are indicated by *, **, or *** for p-values (ANOVA) <0.05, 0.01, or 0.001, respectively.

displayed robust rhythm of *LHY::LUC*, even after 6 days in darkness (**Figure 5D**). The period analysis of *LHY::LUC* expression in *ELF3-Bay* and *ELF3-A362V* revealed no statistically significant period difference in DD, whereas under LL, *ELF3-A362V* displayed a shorter period compared to *ELF3-Bay* (**Figure 5E**). These results are consistent with those described above for both HIFs and NIL-S and

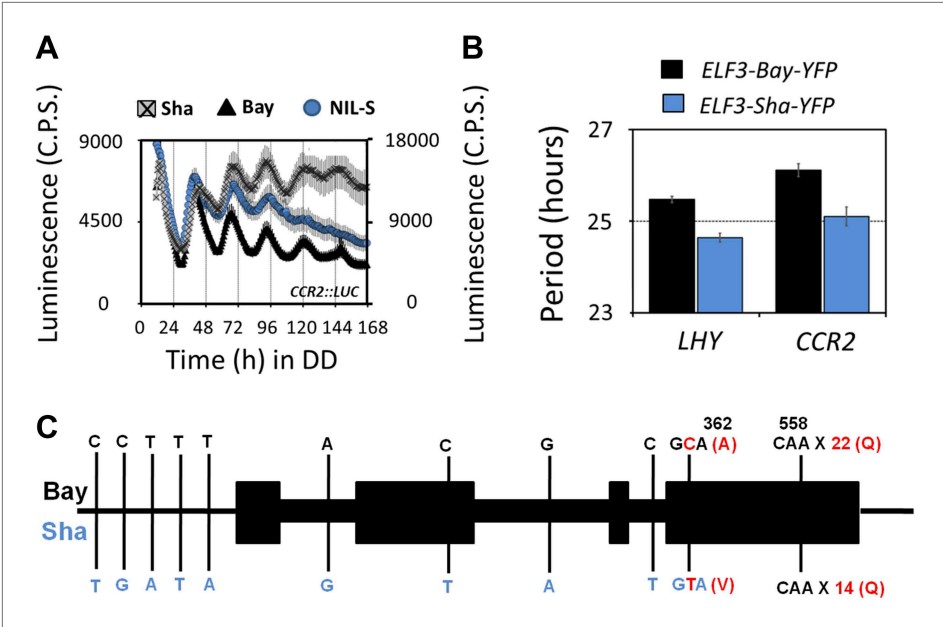

**Figure 3**. Transgenic complementation confirms *ELF3* as QTG underlying Chr2 QTL. Free-running profile of *CCR2::LUC* expression in Bay-0, Sha, and NIL in darkness (**A**), Left y-axis shows the luminescence measures of Bay and NIL, and the right y-axis shows the luminescence measures of Sha. NIL with introgression of Sha at *ELF3* could not sustain the robust rhythms of *CCR2::LUC* after 4 days in darkness. Error bars represent SEM, n ≥ 24. (**B**) Free-running period estimates of *CCR2::LUC* and *LHY::LUC* expression in T2 transgenic lines harboring either *ELF3-Bay-YFP* or *ELF3-Sha-YFP* in Ws-2 background. The data are the average of three independent, single-insert lines displaying similar rhythm profile. Error bars represent SEM, n = 96. (**C**) Sequence comparison of *ELF3-Bay* and *ELF3-Sha*. Schematic representation of *ELF3* (AT2G25930). Vertical bars show the position of the nucleic acid transition. The letters above the bars represent the nucleic acid in Bay, and letters below represent the nucleic acid in Sha. The numbers above the letters represent the position of the nonsynonymous change. The letters in parenthesis show the amino-acid change.

support that the encoded A362V polymorphism is the functionally encoded variant in Sha sufficient to explain the Chr2 periodicity QTL in the BxS population (*Figure 2A,B*). Thus, we could explain the molecular basis of a clock periodicity QTL to an absolute level of a single nucleotide, colloquially defined as the 'quantitative trait nucleotide' (QTN).

## Alanine362 is a required residue to sustain robust rhythmicity in darkness

To further dissect the effect of light and darkness on *ELF3-Sha*, we measured *LHY::LUC* expression in *ELF3-Bay* and *ELF3-A362V* for 15 days with a defined regime of extended light and darkness. We observed that both *ELF3-Bay* and *ELF3-A362V* displayed robust *LHY::LUC* expression so long as they remained under light (*Figure 6A*). However, when the plants were transferred to darkness, *LHY::LUC* expression dampened more rapidly in *ELF3-A362V*. For *ELF3-Bay*, an acute reduction of *LHY::LUC* expression was seen such that proper rhythm was maintained throughout the dark period, while no detectable rhythmic expression of *LHY::LUC* was observed in *ELF3-A362V* (*Figure 6A*). To test if the loss of rhythm of *LHY::LUC* in *ELF3-A362V* was light-dependent or resulted from a 'permanent' defect in the clock caused by continuous darkness, we subsequently returned these plants to light after an interval of 4 days in darkness. Both *ELF3-Bay* and *ELF3-A362V* recovered robust rhythms of *LHY::LUC* after the 11-day treatment, including the 4-day period in darkness (*Figure 6A*). These results suggested that light is necessary to sustain a robust oscillator in the context of *ELF3-A362V*.

Rhythmic robustness depends on the precision of the circadian clock, and relative amplitude error (R.A.E.) is a measure of such precision (*Kolmos et al., 2009*). Therefore, to test the precision of *LHY::LUC* expression rhythms in *ELF3-Bay* and *ELF3-A362V*, we estimated the period and R.A.E. of these lines under LL and in DD. Under LL, *ELF3-A362V* displayed a ~1.5-hr short period compared to

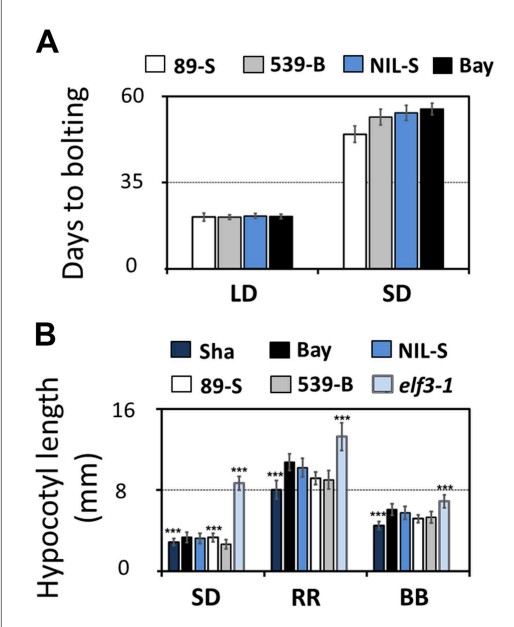

**Figure 4**. Flowering time and hypocotyl length measurements for *ELF3-Sha*. (**A**) Flowering time of HIF 89-S, HIF 539-B, NIL, and Bay-0 under long day (16L:8D) and short day (8L:16D). The flowering time was counted as the number of days at the appearance of 1 cm bolt. (**B**) Hypocotyl length of HIF 89-S, HIF 539-B, NIL, *elf3-1*, Sha, and Bay-0 under short day (8L:16D), under RR (15 μmol m⁻²s⁻¹), and under BB (15 μmol m⁻²s⁻¹). Error bars represent the standard deviation. Significance as described in *Figure 2* compared to Bay-0, except HIF 89-S, which was compared to HIF 539-B.

*ELF3-Bay*, with both lines showing high precision (R.A.E. <0.20; *Figures 6B and 7A*, ZT40-120). However, in darkness, while both lines had similar period estimates, *ELF3-A362V* did not maintain precision and displayed high R.A.E. (R.A.E. >0.75), compared to *ELF3-Bay* (R.A.E. <0.35; *Figures 6B and 7B*, ZT160-240). Interestingly, the loss of precision of *LHY::LUC* rhythm was recovered in *ELF3-A362V* when the plants were returned to the light at ZT250 (*Figures 6B and 7C*, ZT260-340). This restoration of precision led us to hypothesize that in darkness the clock remains partially functional in *ELF3-Sha* in the context of this low precision. To confirm this, we performed an experiment where the phase of *ELF3-Bay* and *ELF3-A362V* lines was estimated after resumption of plants to LL that were previously subjected to different intervals of darkness. We reasoned that if the *ELF3-A362V* plants are truly arrhythmic in DD, then the circadian phase after resumption of light should be determined solely by the exposure to light and not by the duration of the dark period. For *ELF3-Bay*, with a more functional clock in the dark, the phase will be determined in part by the duration of dark exposure. We found that *LHY::LUC* peaked at a similar time in both *ELF3-Bay* and *ELF3-A363V*, which in part was determined by the duration of the dark period. This revealed the presence of a partially functional oscillator during the dark period in both these lines. Further, no significant phase difference between these lines was detected at any time point tested (*Figure 8A*). Interestingly, consistent with the above results (*Figure 6A*), once the plants were transferred back to the light, robust rhythms of *LHY::LUC* were restored (*Figure 8B*). Notably, in a phase–difference graph (Phase *ELF3-Bay* minus Phase *ELF3-A362V*), a consistent pattern of phase oscillations was detected, supporting the existence of a functional oscillator in both *ELF3-Bay* and *ELF3-A362V* that display differences in their respective resetting behavior (*Figure 6C*). This confirms that ELF3-Sha contributes to light-input to the clock, but displays differences in its 'zeitnehmer' entrainment capacity. Taken together, we concluded that *ELF3-Sha* requires light input to maintain precision of the circadian clock; however, darkness does not fully abolish the low-amplitude oscillation in *ELF3-Sha* plants and that entrainment processes appear altered.

## *ELF3-Sha* is defective in proper clock resetting

In Arabidopsis, light input to the circadian clock follows Aschoff's rule who noted that the activity phase shortens in nocturnal organisms exposed to constant light conditions and lengthens in diurnal organisms (*Aschoff, 1979*). These trends were termed alpha compression and alpha expansion, respectively. This process is also known as parametric entrainment (*Aschoff, 1979*; *Johnson et al., 2003*). ELF3, as a repressor of light signaling to the clock, is expected to be involved in such parametric entrainment (*Covington et al., 2001*). In the past, the availability of null mutants precluded testing such a hypothesis. However, the fact that *ELF3-Sha* is a functional allele could be advantageous in determining if *ELF3* is involved in parametric entrainment. To test this, we measured the free-running period of Bay and NIL-S under an array of fluences, including RR, BB, and 'white' (RB) light. Under diverse intensities and qualities of light, we found that NIL-S displayed a shorter period than Bay. Moreover, both Bay and NIL-S followed Aschoff's rule, displaying shortening of periodicity with increase in fluence rate (*Figure 9A–C*). From these results, one of two conclusions may be

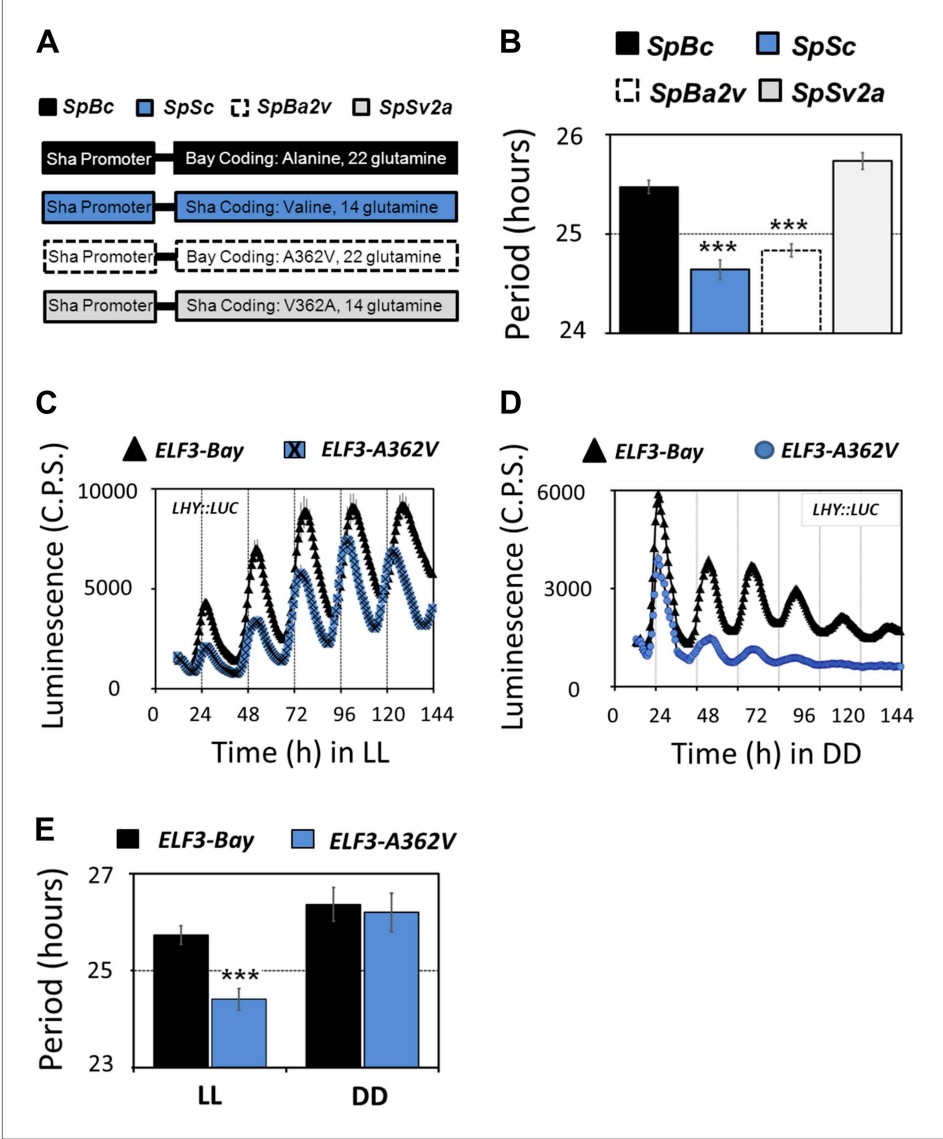

**Figure 5**. Molecular basis of *ELF3-Sha* phenotypes. (**A**) Schematic diagram explaining the different promoter-coding combinations used in (**B**). The Sha promoter of *ELF3* fused with different coding regions is shown (for details see 'Materials and methods'). The encoded amino-acid residue at position 362, along with the number of encoded glutamines, is shown. (**B**) Period estimates of the *LHY::LUC* expression in the lines explained in (**A**). Note that the lines SpSc and SpBa2v with Valine in the coding part displayed period acceleration, irrespective of the number of glutamines. Error bars represent SEM, n = 48. Significance as explained in *Figure 2*, compared to SpBc. Free-running profile of *LHY::LUC* expression in T2 transgenic lines harboring either *ELF3-Bay* or *ELF3-A362V* in Ws-2 genetic background under LL (**C**) and in DD (**D**). A single nucleotide exchange was induced in *ELF3-Bay* to change the encoded alanine residue at position 362 to valine (*ELF3-A362V*). The data are the average of three independent single-insert lines with similar rhythm profile. (**E**) Period estimates of the lines shown in (**C**) and (**D**). Error bars represent SEM, n = 96. Mean values that are significantly different from Bay-0 wild type are indicated by *, **, or *** for p-values (ANOVA) <0.05, 0.01, or 0.001, respectively.

drawn: (1) that ELF3 is not directly involved in parametric entrainment or (2) that the *ELF3-Sha* allele is fully functional in discriminating between different light intensities.

An important characteristic of clock entrainment is termed frequency demultiplication, which describes the resetting properties of the oscillator. Wild-type plants with a functional clock entrained under 24-hr environmental cycles (T24) maintain the same rhythms when transferred to shorter cycles (T12). In this way, plants resist the resetting of the oscillator by sporadic environmental changes during

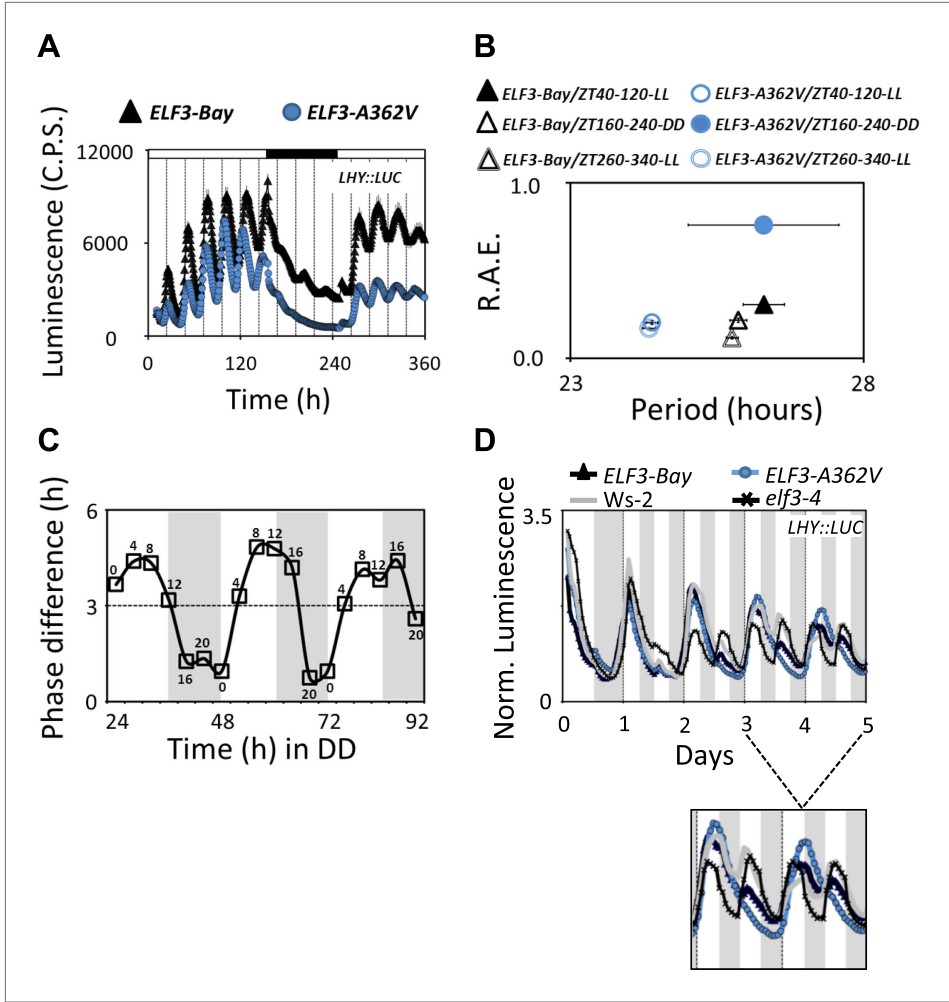

**Figure 6**. Alterations in the *ELF3-Sha* oscillator and clock resetting. (**A**) Free-running profile of *LHY::LUC* expression in *ELF3-Bay* and *ELF3-A362V* in a 15-day continuous experiment under consecutive light and in dark conditions. The plants were entrained for 7 days under 12 hr:12 hr light dark cycles, followed by transfer to LL and measurement of *LHY::LUC* expression for 6 days. On day 7, plants were transferred to darkness, and the measurement of *LHY::LUC* was continued in DD for four more days. On day 11, the plants were transferred to light conditions again, and the expression profile of *LHY::LUC* was measured for an additional 4 days. Open bars in the graph represent time in LL, and closed bar represents time in DD. Error bars represent SEM and are shown on every third reading. (**B**) Period and R.A.E. analysis of profiles shown in (**A**) n = 48. (**C**) Phase shifts in dark-adapted seedlings after resumption to LL. *ELF3-Bay* and *ELF3-A362V* plants entrained for 7 days under light/dark cycles (LD) were transferred in DD for 1 day and then replicate samples were released into LL at 4-hr intervals, monitoring the phase of *LHY* expression in LL to determine the state of the oscillator in the preceding DD interval. Phase difference plot (Phase *ELF3-Bay–ELF3-A362V*) for 3 days in DD is shown. Third peak under LL was used for phase analysis. n = 36. Experiment was repeated three times with similar results. (**D**) Frequency demultiplication assay. After 7 days of entrainment under 12L:12D (T = 24) cycles, the *LHY:LUC* profile was monitored under 12L:12D (T = 24) for 1 day and then 6L:6D (T = 12) for 4 days. The shaded boxes indicate the duration of the LD cycles. For clarification, the *LHY::LUC* profiles from day-3 to day-5 is magnified and shown below.

a diurnal cycle. However, if the short cycles (T12) persist, a robust clock should reset to the new environment. Conversely, when transferred from T24 to T12, mutants with a dysfunctional clock promptly follow the shorter cycles (*Nozue et al., 2007*; *Thines and Harmon, 2010*; *Kolmos et al., 2011*). To test the demultiplication ability of *ELF3-Sha*, we monitored *LHY::LUC* expression for 7 days in *ELF3-Bay*, *ELF3-A362V*, Ws-2, and *elf3-4* lines under T12 cycles (6 hr light and 6 hr dark) after initially entraining under T24 cycles (12 hr light and 12 hr dark). As previously reported for null *elf3* (*Kolmos et al., 2011*), we detected a driven 24 hr rhythm, and, thereafter, *LHY::LUC* completely followed the T12

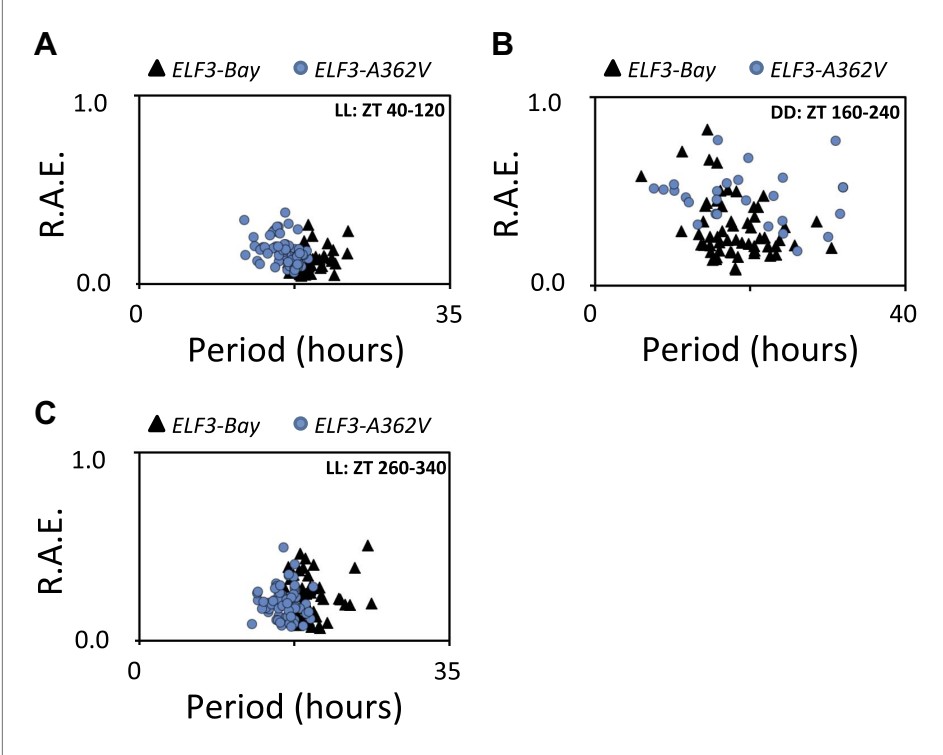

**Figure 7**. Alanine sustains robust oscillator in darkness. (**A**–**C**) Scatter plot for R.A.E. against period showing estimates of individual *ELF3-Bay* and *ELF3-A362V* lines shown in (**Figure 5A**). Only lines with R.A.E. < 1.0 were plotted.

cycles in this mutant, revealing, in turn, a dysfunctional clock (**Figure 6D**). As expected, both Ws-2 and *ELF3-Bay* initially resisted the T12 cycles for 3 days, displaying driven T24 rhythms. However, after 3 days, both Ws-2 and *ELF3-Bay* showed a robust adaptation to the T12 cycles, which were observed as strong *LHY::LUC* peaks matching the peaks expected for T12 cycles (**Figure 6D**). Interestingly, *ELF3-A362V* did not show this effect in T12 cycles, and strong *LHY::LUC* peaks completely matched the T24 cycles for all 5 days (**Figure 6D**). Since both Ws-2 and *ELF3-Bay* responded to persistent T12 cycles, while *ELF3-Sha* did not, we concluded from these results that *ELF3-Sha* failed to perceive persistent new environmental signals, revealing that the circadian oscillator in *ELF3-Sha* is indeed defective in proper entrainment resetting.

## Compromised clock network in *ELF3-Sha*

Under continuous light and in darkness, it has been reported that *elf3* loss-of-function alleles display arrhythmia (**Hicks et al., 1996**; **Thines and Harmon, 2010**). Accordingly, expression profiling revealed that *elf3* had reduced expression of the core oscillator genes *CCA1* and *LHY*, but high levels of the evening genes *TOC1* and *GI* (**Fowler et al., 1999**; **Kikis et al., 2005**; **Dixon et al., 2011**). As these null *elf3* studies were conducted in the context of arrhythmia, placing *ELF3* in the clock network has been difficult. However, as *ELF3-Sha* displayed rhythmicity (**Figure 5C**), we monitored the luciferase reporter expression profile of the central clock genes *CCA1*, *LHY*, *TOC1*, *GI*, *PRR7*, and *PRR9* in NIL-S and compared it with Bay-0 wild type and the null mutants *elf3-1* and *elf3-4* under both LL and DD. Consistent with previous reports (**Kikis et al., 2005**; **Thines and Harmon, 2010**; **Dixon et al., 2011**; **Kolmos et al., 2011**; **Herrero et al., 2012**), both null mutants, *elf3-1* and *elf3-4*, displayed arrhythmia with reduced levels of *CCA1::LUC* and *LHY::LUC* and high levels of *TOC1::LUC*, *GI::LUC*, *PRR7::LUC*, and *PRR9::LUC*, compared to Bay-0 (**Figure 10**). Consistent with the short-period phenotype of *CCR2::LUC* in NIL-S (**Figure 2B**), all clock genes also displayed a short-period phenotype (**Figure 10A–F**). The comparison of expression profiles of these clock genes in NIL-S, *elf3-1*, *elf3-4* and Bay-0 revealed an intermediate effect of *ELF3-Sha*. Specifically, in NIL-S, the expression of *CCA1::LUC* and *LHY::LUC*

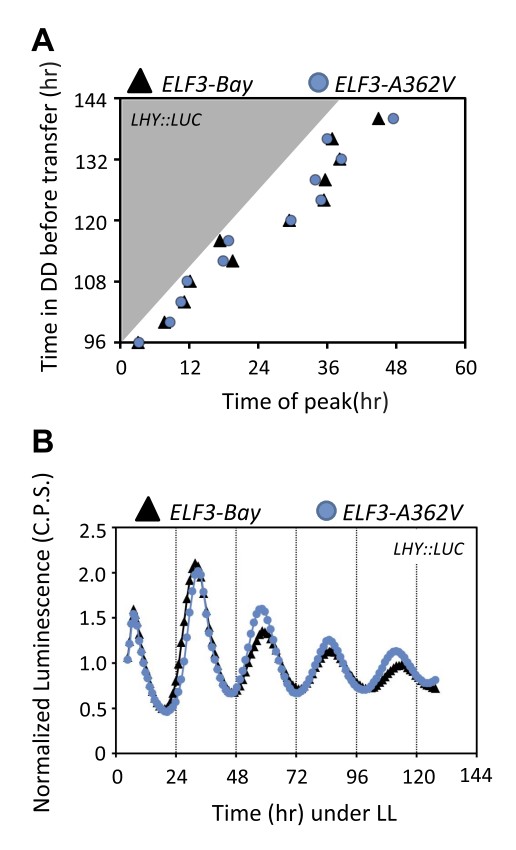

**Figure 8**. Circadian oscillator does not abolish in *ELF3-Sha* in darkness. (**A**) Phase shifts in dark-adapted seedlings after resumption to LL. *ELF3-Bay* and *ELF3-A362V* plants entrained for 7 days under light/dark cycles (LD) were transferred in DD for 1 day and then replicate samples were released into LL at 4-hr intervals, monitoring the phase of *LHY* expression in LL to determine the state of the oscillator in the preceding DD interval. Phase was calculated relative to dawn (ZT00), n = 36. (**B**) The *LHY::LUC* profile in *ELF3-Bay* and *ELF3-A362V* after 128 hr in darkness. Note the robust rhythms of *LHY::LUC* in both lines.

was higher than that of *elf3-1* and *elf3-4*, but lower than that of Bay-0. Similarly, the expression of *TOC1::LUC*, *GI::LUC*, *PRR7::LUC*, and *PRR9::LUC* in NIL-S was lower than that of *elf3-1* and *elf3-4*, but higher than that of Bay-0 (*Figure 10A–F*). Similar to LL, profiles of luciferase expression for clock genes with reduced levels in DD were observed in NIL-S, *elf3-1*, *elf3-4* and Bay-0, except that of *TOC1::LUC* in NIL-S, which was high and comparable to the null mutants (*Figure 10G–L*). A continuous increase in the expression of *GI::LUC* in NIL-S was also observed. No significant period difference was observed in NIL-S and Bay-0 in any clock gene under DD. Consistent results were also obtained when we confirmed the luciferase expression data by monitoring the transcript profile of these genes under LL (*Figure 11*). Unlike *elf3-1* and *elf3-4* loss-of-function mutants, our cumulative results indicate that *ELF3-Sha* is a hypomorphic allele capable of sustaining the oscillation network.

## Cellular basis of *ELF3-Sha* short-period phenotype

A higher expression of morning genes, *PRR7* and *PRR9*, and evening genes, *TOC1* and *GI*, suggests a repressive role of *ELF3*. Thus, the short-period phenotype of *ELF3-Sha* could be caused by a reduced level of *ELF3* transcript or protein accumulation. To test this, we first measured the transcript accumulation of *ELF3* and then assessed the amount of protein generated. We found that the mean level of *ELF3-Sha* transcript was slightly higher relative to *ELF3-Bay*. Correspondingly, higher ELF3-Sha protein accumulation was detected in comparison to *ELF3-Bay*, as measured by the ELF3-YFP signals (*Figure 12A–D*). As increases in ELF3 drives a long period (*Covington et al., 2001*; *Herrero et al., 2012*), elevated protein levels in *ELF3-Sha* thus did not explain the short-period phenotype. Importantly, a more detailed comparison of ELF3-Bay-YFP and ELF3-Sha-YFP revealed differences in cellular localization. Specifically, the formation of distinct nuclear bodies, a characteristic of ELF3 (*Herrero et al., 2012*), was markedly attenuated in ELF3-Sha, whereas these nuclear bodies were clearly observable in ELF3-Bay (*Figure 12A,B*). Additionally, the preferential nuclear localization of ELF3 was compromised in ELF3-Sha, which displayed a markedly elevated amount of cytoplasmic content, compared to ELF3-Bay (*Figure 12A–D*). Quantification of YFP revealed that the nuclear–cytoplasmic ratio of ELF3-Sha was four times less than that of ELF3-Bay (*Figure 12E*). As such, we proposed that these localization defects of ELF3-Sha underlie its oscillator defects.

Having shown that ELF3-Sha is defective in proper cellular localization, we next sought to identify if the localization aberrations cause faulty regulation of ELF3-Sha. This is particularly relevant as it has been shown that ELF3 accumulates under light, but quickly dissipates in darkness by proteasomal machinery involving the action of COP1 and GI. Furthermore, both these proteins physically interact with ELF3 to form similar nuclear bodies that are presumed to be their point of interaction (*Yu et al., 2008*). As such, it could be anticipated that the lack of formation of such nuclear bodies in ELF3-Sha might result in its dysregulation. To examine such a possibility, we monitored the patterns of *ELF3*

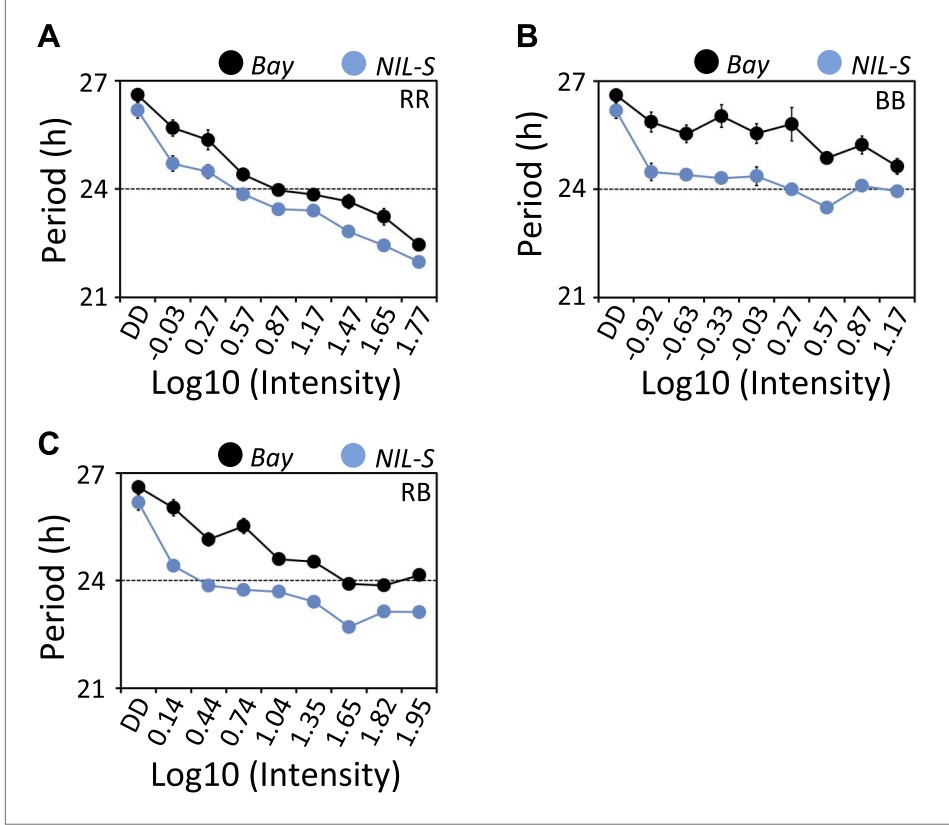

**Figure 9**. *ELF3-Sha* is short-period under a range of light intensities. Free-running period of Bay and NIL-S under different intensities of (**A**) red (**B**) blue and (**C**) red+blue lights (~3:1). Plants were entrained for 7 days under 12 hr LD cycles (white light) before transferring to the respective light conditions. Neutral density filters were used to control the light intensities. Light intensities were measured in μmol m$^{-2}$s$^{-1}$ and were transformed to log10 scale shown in the graphs. DD represents darkness. Error bars represent SEM, n = 36.

transcript and protein accumulation over a diurnal cycle under LD compared to the same patterns under LL and in DD. Under LD, we found only a minor increase in *ELF3-Sha* transcript and protein levels compared to *ELF3-Bay* (*Figure 13A,B*). However, under LL, the *ELF3-Sha* transcript was significantly higher at all time-points compared to *ELF3-Bay*, which also resulted in elevated levels of ELF3-Sha protein (*Figure 13C,D*). Furthermore, compared to LD where ELF3 accumulation decreased during the evening hours, we found elevated protein levels of both ELF3-Bay and ELF3-Sha during subjective night (ZT16 and ZT20). These results were consistent with previous reports that ELF3 accumulation increases in light (*Liu et al., 2001*; *Yu et al., 2008*). In DD, similar to LL, *ELF3-Sha* transcript levels were higher compared to *ELF3-Bay*. However, we did not detect any significant difference in ELF3-Bay and ELF3-Sha protein levels during subjective day, whereas during subjective night (ZT16 and ZT 20), we found that ELF3-Sha accumulation was considerably lower than ELF3-Bay (*Figure 13E,F*). Taken together, a higher accumulation of ELF3-Sha under LL and overall lower levels in DD, despite higher transcript abundance, led us to conclude that ELF3-Sha protein dissipates in darkness comparatively more rapidly than ELF3-Bay.

## Allelic diversity at the *ELF3* locus

Finally, we sought to identify the evolutionary history of the *ELF3-Sha* locus. To accomplish this, we retrieved the *ELF3* coding sequence (CDS) of 251 accessions from the '1001 Genome-Project' ('Materials and methods') and performed several population-genetics analyses. The sequence comparison revealed the presence of 59 polymorphic sites segregating at the *ELF3* locus, including encoded A362V. Out of these 59 polymorphic sites, 16 were synonymous and 43 were nonsynonymous. The A362V was detected in 15 accessions, and in a phylogenetic analysis, all of these grouped

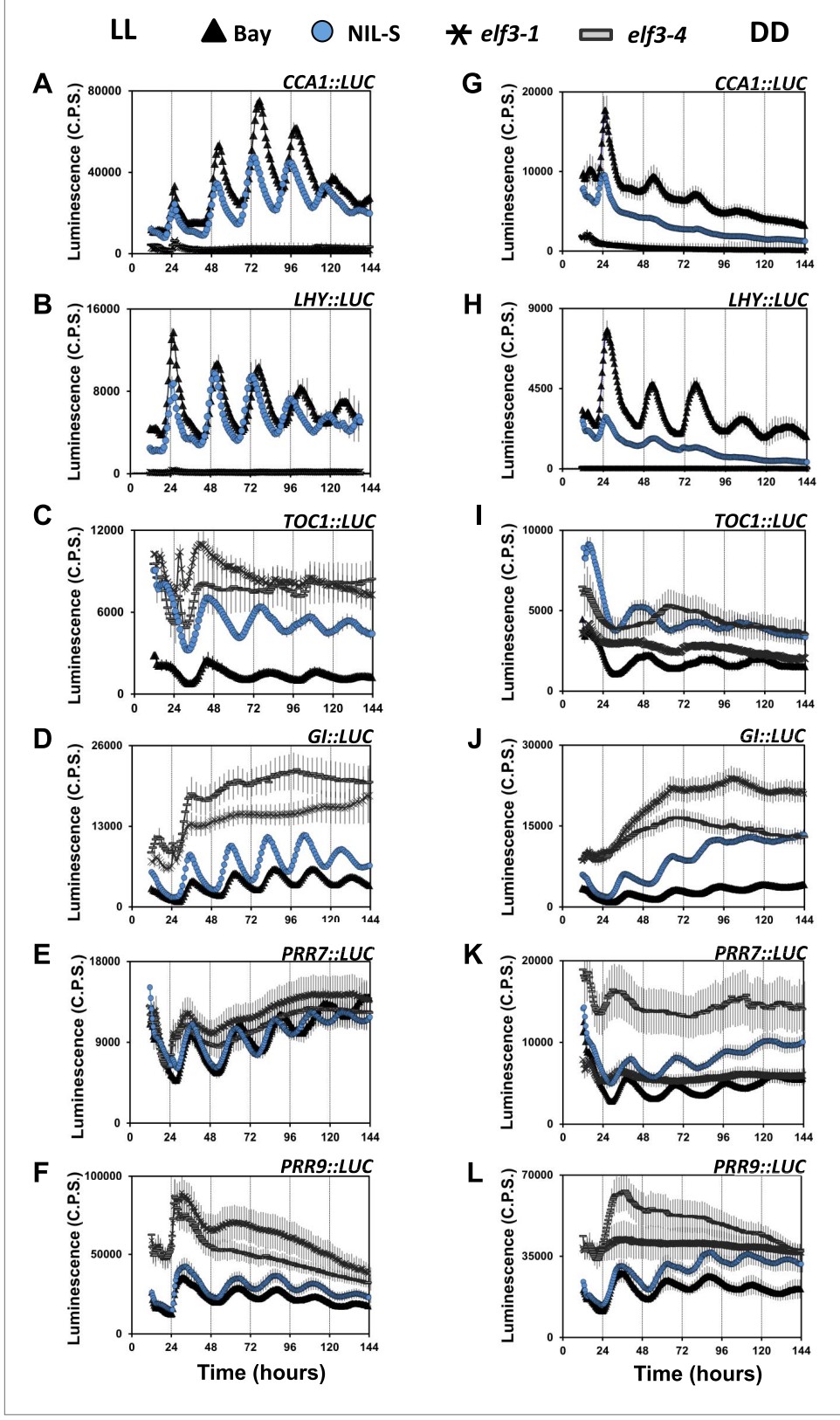

**Figure 10**. Compromised clock network in *ELF3-Sha*. Luciferase expression profile of different clock genes in NIL, *elf3-1*, *elf3-4* and Bay-0 under LL (left panel, **A**–**F**) and in DD (right panel, **G**–**L**). (**A** and **G**) *CCA1::LUC*, (**B** and **H**)

*Figure 10. Continued on next page*

*Figure 10. Continued*

*LHY::LUC*, (**C** and **I**) *TOC1::LUC*, (**D** and **J**) *GI::LUC*, (**E** and **K**) *PRR7::LUC*, and (**F** and **L**) *PRR9::LUC*. Error bars represent SEM and are shown on every third reading. Note that the NIL-S displayed an intermediate expression of all clock genes relative to the null mutants *elf3-1* and *elf3-4* compared to Bay-0.

together in a single clade, suggesting a common origin (***Figure 14A***). Therefore, we looked at the geographical distribution of these accessions and found that they all were distributed in Central Asia between 37°N and 54°N, with two exceptions at Nemrut and Rubenzhoe collected from Turkey and the Ukraine, respectively (***Figure 14B***; ***Table 4***). Interestingly, similar to Sha that naturally grows at high altitudes (e.g., Pamir Mountains, Tajikistan), most of these accessions were also high-altitude accessions collected in the mountains (***Table 4***). These results suggest that *ELF3-Sha* originated from Central Asia and that this allele might have been maintained during species migration preferentially by altitude-associated individuals.

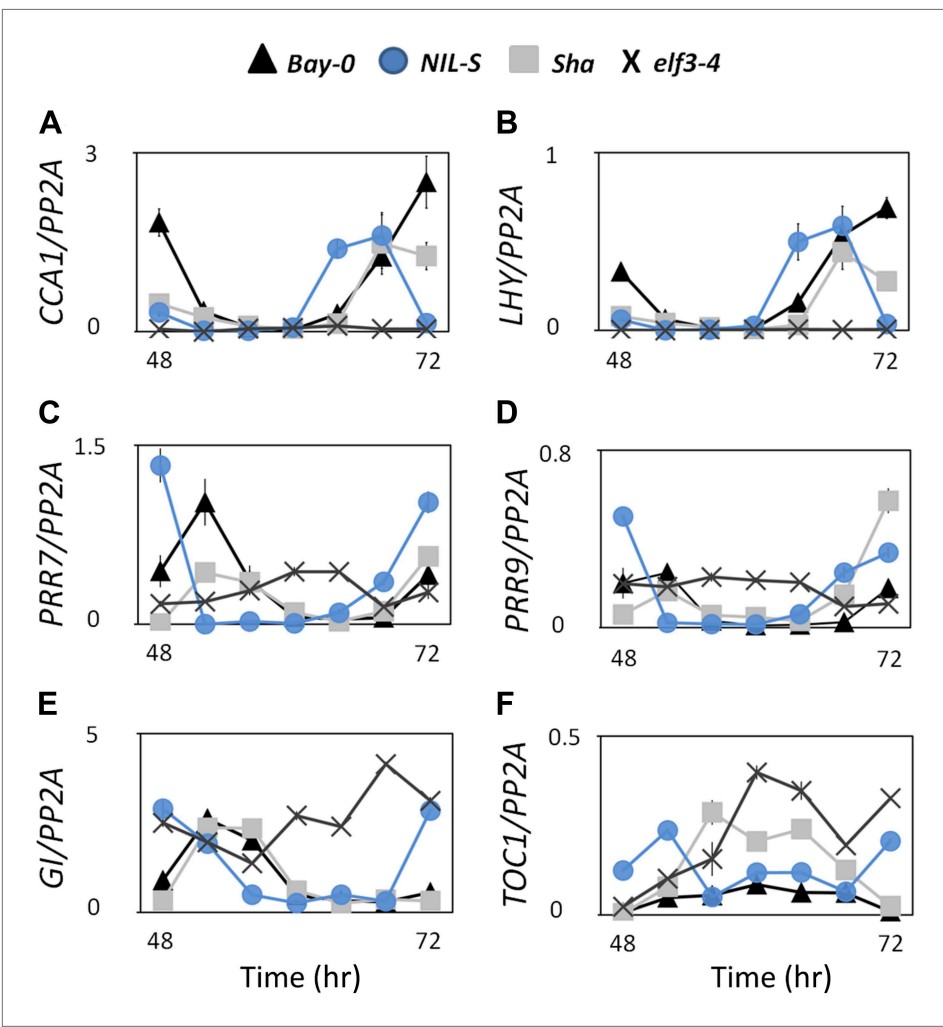

**Figure 11**. Transcript accumulation pattern of different clock genes under LL. Transcript accumulation of different clock genes in Bay-0, Sha, NIL-S and *elf3-4* under LL. (**A**) *CCA1::LUC*, (**B**) *LHY::LUC*, (**C**) *PRR7::LUC*, and (**D**) *PRR9::LUC*, (**E**) *GI::LUC*, and (**F**) *TOC1::LUC*. Error bars represent the standard deviation of three technical repeats. Expression levels are normalized for *PROTEIN 19 PHOSPHATASE 2a subunit A3* (*PP2A*). Growth conditions, quantitative RT-PCR, and primer sequences were previously described (***Kolmos et al., 2009***; ***Kolmos et al., 2011***).

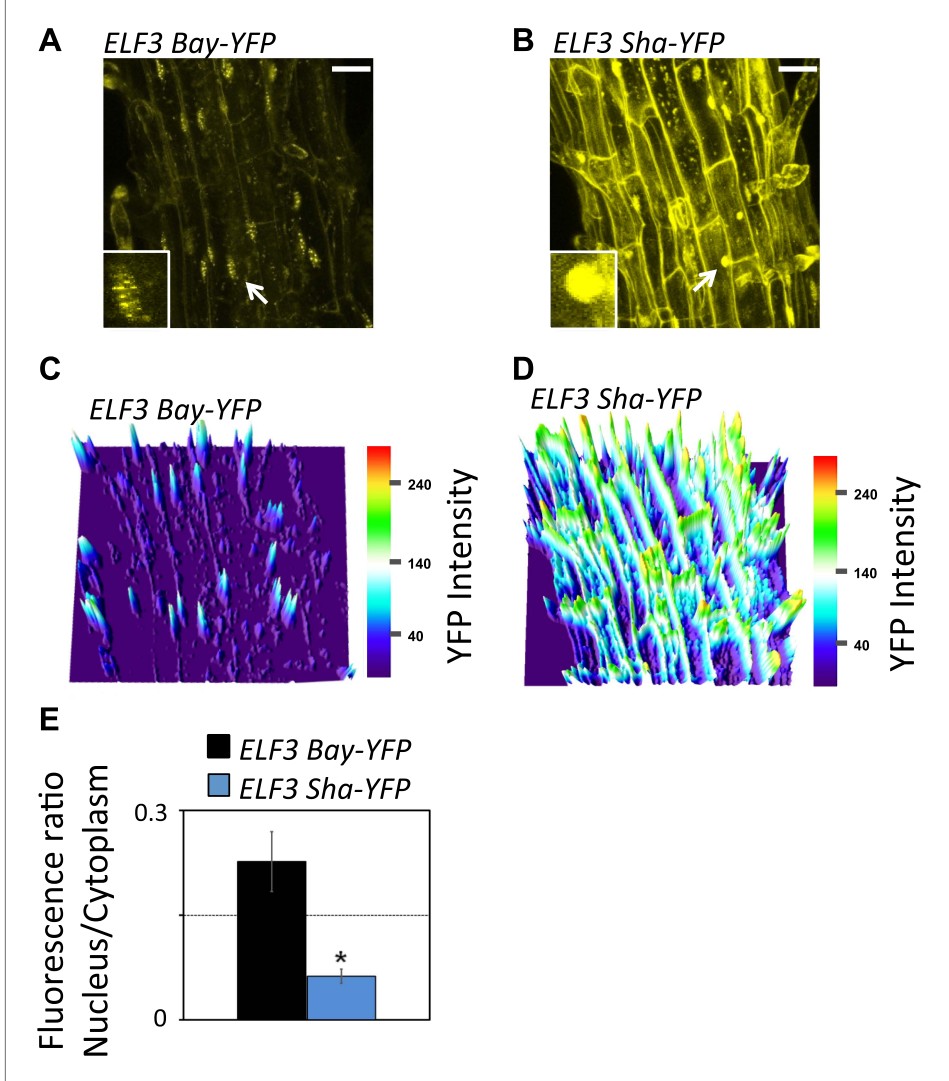

**Figure 12**. Sub-cellular localization defects of ELF3-Sha. (**A**) and (**B**) show maximum intensity projection of ELF3-YFP localization in root cells of ELF3-Bay-YFP (**A**) and ELF3-Sha-YFP (**B**). Arrows indicate the nuclei that are magnified four times and shown in small boxes at the bottom of (**A**) and (**B**). Note that ELF3 forms distinct nuclear foci in ELF3-Bay, whereas in ELF3-Sha, YFP signal for ELF3 is diffused in the nucleus. Scale bar is 20 μm. (**C**) and (**D**) display the YFP intensity distribution of (**A**) and (**B**) in visual-thermal units, respectively. Note that the ELF3 cytoplasmic contents were higher in ELF3-Sha as compared to ELF3-Bay. (**E**) shows the nucleus-to-cytoplasmic fluorescence ratio of ELF3-Bay-YFP and ELF3-Sha-YFP, as calculated by ImageJ. Error bars represent SEM, n = 3. Significance as described in *Figure 2*. The representative data of three independent experiments and three independent lines are shown.

## Discussion

*ELF3* has been established as a required component for the generation of circadian rhythms and the perception of light and temperature inputs (*Covington et al., 2001*; *Thines and Harmon, 2010*; *Kolmos et al., 2011*). However, most of the *elf3* alleles used in these studies were identified in mutagenesis screens, and many of them displayed arrhythmicity under free-running conditions of LL and DD. Here, we report the cloning and characterization of *ELF3-Sha* as a natural allele of *ELF3*, which displayed a light-dependent short-period phenotype. Unlike *elf3* loss-of-function mutants, this allelic state displayed robust circadian rhythms under continuous light. However, in prolonged darkness, the expression profile of many clock genes was dramatically damped (*Figure 10*). The association between the clock phenotypes of *ELF3-Sha* and their cellular basis led us to conclude that proper cellular

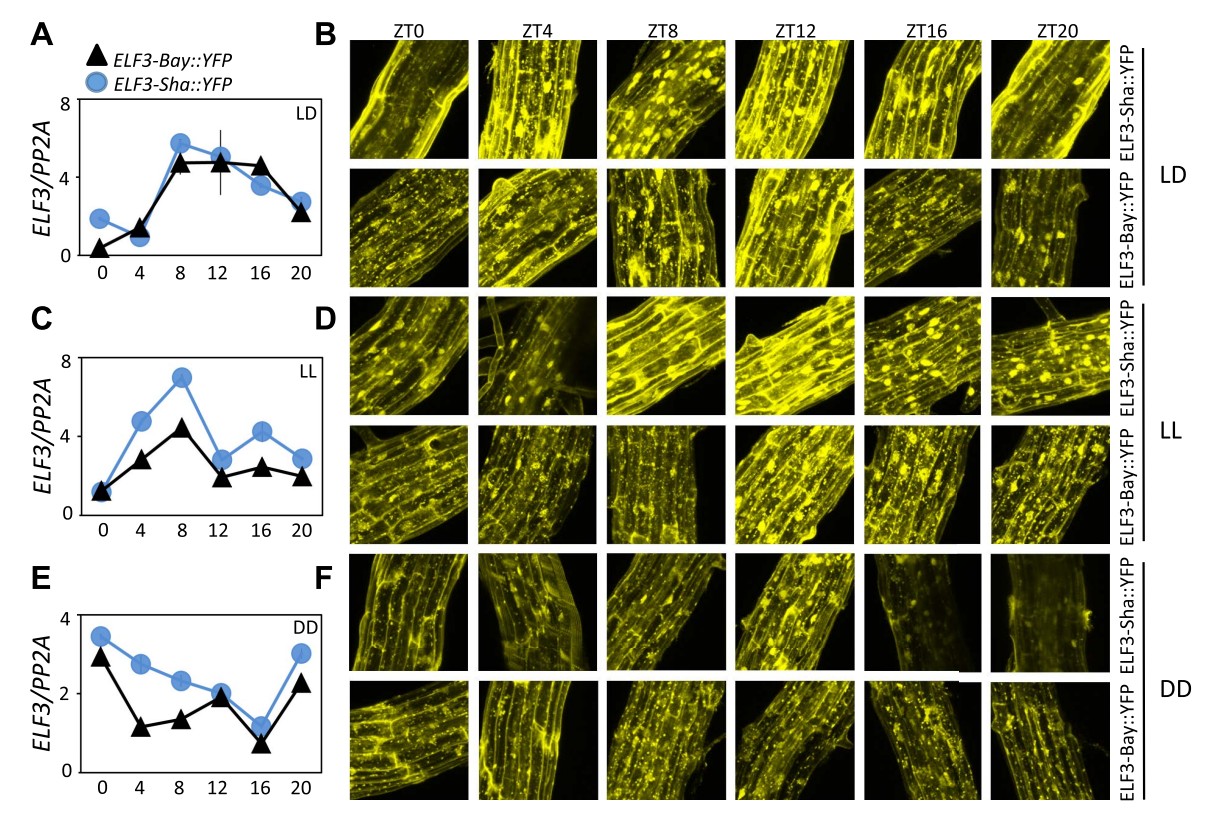

**Figure 13**. ELF3 cyclic accumulation is altered in *ELF3-Sha*. Accumulation pattern of *ELF3* transcript and encoded protein in *ELF3-Bay-YFP* and *ELF3-Sha-YFP* lines under LD (**A** and **B**), LL (**C** and **D**) and in DD (**E** and **F**). For LD, plants were grown under 12L:12D cycles for 6 days, and starting the next day at ZT0, plants were harvested every 4 hr for RNA extraction and then, separately, scanned under the microscope for cellular studies. For LL and DD, after initial entrainment, plants were transferred under white light or in darkness for 1 day, followed by harvesting the samples for RNA extraction or taking pictures for their respective zeitgeber time. This microscopic experiment was performed three times with similar results.

distribution of ELF3 is required to perform normal repressive function in the circadian clock (*Figures 12 and 13*) which in turn, depends on its cyclic accumulation. Importantly, we could track the origin of *ELF3-Sha* to Central Asia, providing more insight about the molecular evolution of this allele.

The identification and validation of quantitative trait loci (QTLs) by genetic mapping is a well-established procedure. However, to understand the molecular basis of a locus by characterizing QTL to a quantitative trait nucleotide (QTN) level is still an arduous task (*Koornneef et al., 2004*). Here, we reported the characterization of a natural allele of *ELF3 (ELF3-Sha)* to a QTN level of understanding. In the quantitative analysis of circadian periodicity of a modified BxS population, we identified a QTL and further validated it in HIFs and a NIL (*Table 1*; *Figure 2A,B*). Our periodicity results were consistent with previous studies where the *ELF3* in Sha was found to modulate the shade-avoidance response, including the action of shade on the oscillator (*Jimenez-Gomez et al., 2010*; *Coluccio et al., 2011*). In those studies, HIFs were the main genetic resource used. The authors of those papers proposed *ELF3* to be a candidate locus for the detected shade-avoidance QTL, and transgenic efforts were used to complement that notion (*Jimenez-Gomez et al., 2010*). Here, we generated the appropriate NIL and showed that the introgression of *ELF3-Sha* to Bay-0 resulted in severe circadian alterations, including a curtailed period length (*Figure 2B*).

The light-dependent acceleration of circadian oscillations of *ELF3-Sha* could be associated with reduced ELF3 functionality (*McWatters et al., 2000*; *Covington et al., 2001*; *Kolmos et al., 2011*). The intermediate expression profile of several clock genes in *ELF3-Sha,* when compared to wild-type and null mutants, confirmed that it is a hypomorphic allele (*Figures 10 and 11*). This characteristic of *ELF3-Sha* is distinctly different from *elf3* null mutants, which displayed complete arrhythmia under LL (*Covington et al., 2001*; *Hicks et al., 2001*; *Thines and Harmon, 2010*; *Kolmos et al., 2011*).

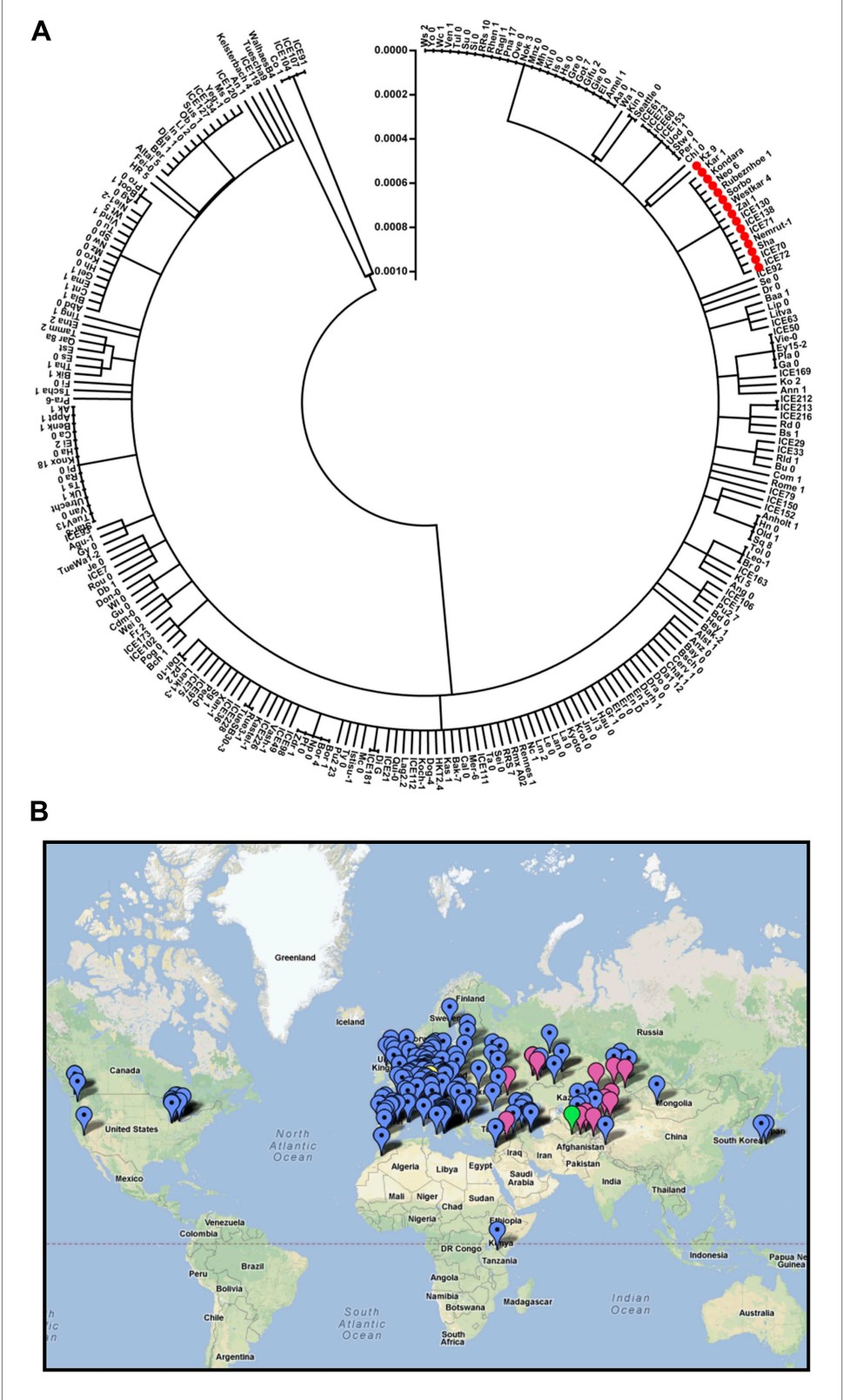

**Figure 14**. Distribution of the *ELF3* locus. (**A**) Neighbor-joining tree showing the phylogenetic relationship of *ELF3* coding sequence of 251 accessions. All accessions harboring *ELF3-Sha* allele were grouped in a single clade indicated by red-filled circles. Scale represents the distance calculated by interior–branch test. Sites with gaps/missing
*Figure 14. Continued on next page*

*Figure 14. Continued*

data were not included in the analysis. (**B**) Geographical distribution of *ELF3-Sha* allele. Accessions harboring *ELF3-Sha* are shown with pink marks, whereas blue marks represent *ELF3-Bay* allele. Locations of Bay and Sha are shown with yellow and green marks, respectively.

However, the functionality of the oscillator in these mutants in darkness remained disputable. Based on the observation that the circadian rhythms persist in the *elf3-1* mutant, it was proposed that ELF3 solely acts as a light *zeitnehmer*, or timekeeper, and that it is not a required component of the oscillator. In contrast, a recent study reported that the circadian oscillator was abolished in constant darkness (DD) in temperature-entrained *elf3-1* seedlings, emphasizing that ELF3 is an integral part of the oscillator as well as acting as a *zeitnehmer* (**Thines and Harmon, 2010**). As these studies were mainly based on null mutants, a conclusive statement about ELF3 function in sustaining the oscillator in DD is hard to draw. Consequently, in this paper, we thus analyzed the profile of several clock genes in the functionally stable *ELF3-Sha* mutant in DD, comparing it with those in *elf3-1* and *elf3-4* null mutants. Consistent with a previous report (**Thines and Harmon, 2010**), we could not detect any sign of rhythmicity in null *elf3* mutants. Interestingly, in the hypomorphic *ELF3-Sha*, a rapid dampening in the expression of several clock markers in DD was observed (**Figures 3A, 5D, 6A**, and right panels of **Figure 10G–L**). Thus, our results conclusively demonstrate that ELF3 is required for robust oscillation in darkness and further support the idea that ELF3 is an integral part of the oscillator (**Thines and Harmon, 2010**). Although *ELF3-Sha* lost robustness and precision in prolonged darkness, it is important to note that it quickly recovered when transferred back to the light (F**igures 6A,B, 7A–C**). Consequently, we further propose that the function of *ELF3* is collectively defined by both the light–dark boundary and by circadian clock regulation.

   *elf3* loss-of-function mutants under free-running conditions were previously shown to display major defects in the expression of the core oscillator genes *PRR7*, *PRR9*, *CCA1*, *LHY*, *TOC*1, and *GI*

**Table 4.** Geographical location of the 15 accessions with *ELF3-Sha* allele

| Abbreviation | Name | Stock ID | Country | Location | Latitude | Longitude | Altitude |
|---|---|---|---|---|---|---|---|
| ICE130 | Kolyvan/Kly-4 | CS76384 | Russia | Kolyvan | 51.32 | 82.55 | 505 |
| ICE138 | Lebjashje/Leb-3 | CS76426 | Russia | Altaijskij Kraj | 51.65 | 80.82 | 301 |
| ICE70 | Borskoje/Borsk-2 | CS76421 | Russia | Samarskaja Oblast | 53.04 | 51.75 | 75 |
| ICE71 | Shiguljovsk/Shigu-1 | CS76375 | Russia | Samarskaja Oblast | 53.33 | 49.48 | 181 |
| ICE72 | Shiguljovsk/Shigu-2 | CS76374 | Russia | Samarskaja Oblast | 53.33 | 49.48 | 181 |
| Kar1 | Karakol | CS76522 | Kyrgyzstan | Susamyr village/West Karakol river | 42.3 | 74.36 | 2503 |
| Kondara | Kondara | CS76532 | Tadjikistan | Khurmatov | 38.5 | 68.5 | 815 |
| Kz9 | Kazakhstan | CS76537 | Kazakhstan | Karagandy | 49.67 | 73.33 | 535 |
| Nemrut | Nemrut-1 | CS76398 | Turkey | Nemrut Dag | 38.64 | 42.23 | 2249 |
| Neo6 | Neo6 | CS76560 | Tajikistan | Jawshangoz village | 37.35 | 72.46 | 3467 |
| Rubenzhoe | Rubenzhoe-1 | CS76594 | Ukraine | Rubezhnoe | 49.01 | 38.36 | 56 |
| Sha | Shakdara | CS76382 | Tadjikistan | Pamiro-Alay | 38.90 | 69.05 | 3439 |
| Sorbo | Sorbo | CS22653 | Tadjikistan | | 38.82 | 69.48 | 1751 |
| Westkar4 | West Karakol | CS76629 | Kyrgyzstan | Karakol valley | 42.3 | 74 | 2187 |
| Zal1 | Zal1 | CS76634 | Kyrgyzstan | Tchong-Kemin valley/Djachyl-Kul lake | 42.8 | 76.35 | 2230 |

(*Kikis et al., 2005*; *Dixon et al., 2011*; *Kolmos et al., 2011*). To monitor the expression of these clock genes in *ELF3-Sha*, we used the robust luciferase reporter-expression system and measured these expression profiles in Bay-0, NIL-S, and the nulls *elf3-1* and *elf3-4*. Our results were consistent with previous reports showing the arrhythmic low levels of *CCA1* and *LHY* and high levels of *TOC1* and *GI* in *elf3* loss-of-function mutants. In contrast, under LL, *ELF3-Sha* was rhythmic for all genes studied, and it displayed an intermediate level of expression compared to wild type and loss-of-function *elf3* (*Figure 10A–F*, left panels). Specifically, NIL-S displayed higher expression of *TOC1* and *GI* compared to Bay-0, but lower expression than *elf3-1* and *elf3-4*. In contrast, expression of *CCA1* and *LHY* was lower in NIL-S, compared to Bay-0, but higher than *elf3-4*. Consistent with the previous finding that CCA1 and LHY regulate *PRR7* and *PRR9* (*Farre et al., 2005*; *Nakamichi et al., 2010*), a higher transcript abundance of *PRR7* and *PRR9* was observed in NIL-S (*Figures 10E,F and 11C,D*). These data, along with recent findings showing the evening complex (*ELF3*, *ELF4*, and *LUX*) directly binds to the *PRR9* promoter to repress transcription, support the repressive action of ELF3 in the core oscillator (*Helfer et al., 2011*; *Nusinow et al., 2011*; *Chow et al., 2012*; *Herrero et al., 2012*). Higher expression levels of the evening genes *TOC1* and *GI*, as well as dramatically dampened oscillations of *ELF3-Sha* in darkness, cannot be simply explained by only considering ELF3 at *PRR9*. Considering the double loss-of-function mutant *cca1-11 lhy-21* is rhythmic, an additional repressive role of ELF3 by targeted degradation of GI is conceptually plausible (*Ding et al., 2007*; *Yu et al., 2008*). Taken together, our data support a hypothesis which holds that the EC-containing ELF3 has more than one entry point in the circadian clock (*Kolmos et al., 2009*, *2011*; *Herrero and Davis, 2012*).

ELF3 is both a cytosolic and nuclear localized protein. It appears to be multifunctional in that it has several binding partners. These include phyB, COP1, ELF4, and GI (*Yu et al., 2008*; *Nusinow et al., 2011*; *Herrero et al., 2012*). Different domains of ELF3 specifically interact with these different proteins. Both phyB and COP1 interact with the N-terminal domain (*Liu et al., 2001*; *Yu et al., 2008*; *Kolmos et al., 2011*), whereas ELF4 and GI interact with the middle domain (*Yu et al., 2008*; *Herrero et al., 2012*). Further, all these proteins co-localize in the nucleus where they form distinct nuclear bodies (*Mas et al., 2000*; *Yu et al., 2008*; *Chen et al., 2010*; *Herrero et al., 2012*). Such nuclear foci could be suspected as interaction points of binding proteins (*Yu et al., 2008*; *Herrero and Davis, 2012*; *Herrero et al., 2012*; *Kim et al., 2013*). Thus, formation of fewer ELF3-nuclear foci from encoded *ELF3-Sha* might be the result of a defect in the binding of one of its interacting proteins. Since the A362V variant is located in the middle domain of ELF3, such a hypothesis could be particularly attributed to either GI and/or ELF4. The rhythmic accumulation of both ELF3 and GI depends upon the activity of COP1 that mediates ubiquitination and targeted degradation of these proteins in dark conditions (*Yu et al., 2008*). It is noteworthy here that ELF3 is essential for this process. In the absence of ELF3, COP1 cannot interact with GI and thus cannot initiate its decay. However, GI is not required for COP1-mediated ELF3 degradation (*Yu et al., 2008*). When these data are taken together, the formation of a COP1-ELF3-GI complex could be considered a plausible active mechanism controlling the rhythmic accumulation of these proteins. Under these conditions, A362V mutation in *ELF3-Sha* would result in attenuated binding affinity with GI, disturbing the balance of the COP1-ELF3-GI complex and, in turn, resulting in the rapid decay of ELF3. Both the lower accumulation of ELF3-Sha protein, despite the generation of higher transcript (*Figure 13E,F*), and aberrant oscillator behavior of ELF3-Sha in prolonged darkness support this hypothesis. The short-period phenotype of *ELF3-Sha* under LL could also be explained by an early decay of ELF3-Sha during light/dark entrainment. As ELF3 directly represses *PRR9* (*Dixon et al., 2011*; *Herrero et al., 2012*), the early depletion of ELF3-Sha results in an early expression of *PRR9*, an event that sets the pace of the oscillator during the preceding entrainment, which is maintained through LL. Based on these results and our observation that *ELF3-Sha* is defective in proper clock resetting (*Figure 6D*), we propose that ELF3 plays a pivotal role in defining entrainment properties of the circadian clock, which, in turn, depend on cyclic accumulation of ELF3 protein, but not on its absolute levels or, indirectly, its transcript abundance. This idea could be further supported by the observation that ELF3 protein rhythmically oscillates and drives a long period in plants that constitutively overexpress *ELF3* (*Covington et al., 2001*; *Dixon et al., 2011*; *Herrero et al., 2012*). Discounting the important role that ELF3 plays in clock entrainment, it can be provisionally excluded that ELF3 is involved in parametric entrainment because we could not detect any differential effect of fluence rate on *ELF3-Sha* periodicity compared to *ELF3-Bay* (*Figure 9A–C*). Fluence rate curves based on *ELF3-ox* also support this notion (*Covington et al., 2001*).

Studies on *ELF3-Sha* as a natural allele disrupted in normal circadian behavior provide a perspective on its repressive action on clock periodicity. Notably, in the context of the Ws-2 and Bay-0 genomes, we could show that *ELF3-Sha* is a hypomorphic allele defective in proper localization of encoded ELF3 protein. This defect resulted in two distinct phenotypes: one displaying a light-dependent short period and the other exhibiting loss of rhythm robustness in darkness. These phenotypes were clearly observable in both the Bay-0 and Ws-2 genetic background. It is notable that Sha parental line did not itself display such obvious circadian defects (*Figures 2B and 3A*). Related to this, the transcript profile of clock genes in Sha followed the same pattern as observed in Bay-0, and such pattern was distinct from that seen in NIL-S (*Figure 11*). Moreover, *ELF3-Sha* did not affect hypocotyl length in NIL-S and Sha in the same way. Under all light qualities tested, the hypocotyl length of NIL-S was similar to that of Bay-0, but was significantly longer when compared to Sha (*Figure 4B*), leading to the possibility that other segregating QTLs within Sha wild type genetically interact with *ELF3-Sha*. Such similar background-dependent effects of natural alleles have been reported in Arabidopsis for seed longevity, axillary bud formation, and flowering time (*Sugliani et al., 2009*; *Huang et al., 2012*; *Undurraga et al., 2012*; *Méndez-Vigo et al., 2013*).

Our efforts to track the history of the *ELF3-Sha* revealed that this allele is present in a group of genetically related accessions that are predominately distributed in a geographical area of Central Asia (*Figure 14A,B*). This confined location of *ELF3-Sha* leads to the possibility that this allele provides local adaptive advantage to these accessions under their respective environmental conditions, and thus, has been positively selected during species migration. Such a hypothesis can only be confirmed in further studies using realistic environmental conditions in which these accessions are derived.

## Materials and methods

### Plant material

The Recombinant Inbred Lines (RILs) used were derived from the Bayreuth-0 (Bay-0) by Shakdara (Sha) RIL collection (termed here BxS) (*Loudet et al., 2002*). Multiple, independent T1 transgenic *CCR2::LUC* reporter lines were obtained from 71 lines after floral dipping (*Boikoglou et al., 2011*; *Davis et al., 2009*). T2 progeny were used for circadian rhythm experiments (*Supplementary files 1 and 2*). To confirm the chromosome 2 periodicity QTL in BxS, three heterogeneous inbred families (HIFs) were generated. For this, RILs 57, 92, and 343 were used as recipients of a pollen donation from Bay-0 *CCR2::LUC*. These F1 lines were respectively backcrossed twice to the given RIL. In these three BC2 populations from the separate RIL crosses, the plants were self-crossed, and in the derived BC2F2 populations, lines that were homozygous for Bay or for the Sha alleles, at four marker positions, were identified. Multiple F2 versions of each of these derived lines were isolated. F3 seeds that harbored *CCR2::LUC* were collected from these plants for periodicity tests, as described below. The selected F3 HIFs containing Sha at QTL interval were further backcrossed three times to Bay-0 to generate NIL-S. The introgression of Sha at QTL interval and the homogeneous Bay-0 background were confirmed with genome-wide SSLP markers. For fine mapping, the progeny of HIF343, heterozygous at QTL interval, was screened with different SSLP, CAPS, and dCAPS markers. Out of 1100 plants screened, 14 plants with a recombination event between markers elf100L and elf100R were selected and self-fertilized to obtain homozygous recombinant lines. The progeny of these homozygous recombinant lines was then used for circadian-periodicity assay. The detail of all the markers used for genotyping is given in *Supplementary file 3*.

The mutant lines used in this study were as follows: *elf3-1* and *elf3-4* (*Zagotta et al., 1992*, *1996*; *Hicks et al., 2001*). Both mutant lines were backcrossed four times to the Bay-0 wild type to homogenize the accession background to Bay-0. Homozygous plants were subsequently identified in the BC4F2 population using specific markers (*Supplementary file 3*). Different clock-marker lines, *CCR2::LUC*, *CCA1::LUC*, *LHY::LUC*, *TOC1::LUC*, and *GI::LUC*, used in the study were generated by initial transformation of the respective marker into Bay-0, followed by crossing the T2 transformants to the target genotype: Bay-0, NIL-S, *elf3-1*, and *elf3-4*. The homozygous lines obtained by the self-fertilization of BC1F2 were used for the circadian assays.

To generate *ELF3 Bay* transgenic lines, the *ELF3* gene, along with *ELF3* native promoter, was amplified from Bay-0 genomic DNA and cloned into pPZP211 vector. In *ELF3 A362V*, a nucleotide change encoding Alanine to Valine (A362V) was induced using the QuikChange method (Stratagene, California, USA). The multi-gateway technology (Invitrogen, California, USA) was used to generate the YFP-tagged

lines with different promoter-coding combinations. Initially, the promoter and coding regions of *ELF3* were separately amplified from Bay-0 and Sha genomic DNA and recombined into pDONR4-P1R or pDONR201 donor vectors, respectively. The respective nucleotide change encoding either the A362V or V362A amino acid was then induced using the QuikChange method (Stratagene). The YFP tag was separately cloned into pDONRp2r-p3 donor vector. These three donor vectors were then recombined in different combinations into pPZP211 destination vector. The final vector was then transformed into *Agrobacterium tumerfaciens* (strain ABI). For all transformations, the improved floral-dip method was used (*Davis et al., 2009*). Based on Mendelian segregation, T2 transgenic lines having single insertion were selected on kanamycin. All the transgenic lines harboring *ELF3-Bay, ELF3-A362V, ELF3-Bay-YFP, ELF3-Sha-YFP, SpSc, SpBc, SpBa2v* and *SpSv2a* are in *elf3-4 Ws-2* genetic background. All oligonucleotides used for cloning and QuikChange are listed in *Supplementary file 3*.

## Plant growth conditions, luciferase assay, and measurement of developmental traits

For luciferase assays, seeds were surface-sterilized and plated on MS medium containing 3% sucrose. Following ~3 days stratification at 4°C, seedlings were entrained for 7 days, either under 12L:12D cycles (~100 μE light) with constant temperature of 22°C (LD) or under 12 hr at 16°C: 12 hr of 22°C temperature cycles with constant light (TMP) (~100 μE light). The bioluminescence measurement and data analysis are as described (*Hanano et al., 2006, 2008*). For hypocotyl assays, seedlings were grown on MS medium (2.2g/L pH 5.7) without sucrose, as described (*Davis et al., 2001*). Hypocotyl length was determined for seedlings grown under SD (8L:16D), BB, or RR for 7 days (light intensity SD: 120 ìmol m-2s-1; light intensity RR and BB: 15 ìmol m-2s-1). Seedlings were scanned, and hypocotyl elongation was measured using the ImageJ 64 program, V1.43b (Wayne Rasband, National Institutes of Health, USA, http://rsb.info.nih.gov/ij). For flowering time measurement, plants were grown on soil containing a 3:1 mixture of substrate and vermiculite in a temperature-controlled greenhouse environment with 16L:8D long-day and 8L:16D short-day cycle. The flowering time was scored at the time of bolting (1 cm above rosette leaves) as the total number of days to bolt (*Domagalska et al., 2010*).

## QTL mapping and analysis

In total, 60 and 65 BxS lines were assayed for *CCR2* rhythmic periodicity after light and temperature entrainment, respectively. Period mean was subsequently used for QTL mapping, which was performed with MapQTL 5.0 (Kyazma BV, Wageningen, The Netherlands). Mapping settings used were as in *Boikoglou et al. (2011)*. The statistical analyses, including broad sense heritability were calculated as reported (*Keurentjes et al., 2007*; *Boikoglou et al., 2011*).

## Confocal microscopy

For all microscopic work, the Zeiss LSM700 confocal microscope from Carl Zeiss was used, as in *Herrero et al. (2012)*. Briefly, the plants were grown on MS medium containing 1.5% sucrose. Following ~3 days stratification at 4°C, seedlings were entrained for 6 days under 12L:12D cycles (~100 μE light) with constant temperature of 22°C. The following day, the plants were either put under constant light (LL: ~100 μE light) or in darkness for another day. On day 7, starting at ZT0, the plants were scanned, and the photographs were taken every 4 hr for 1 day. For the comparison of *ELF3 Bay-YFP* and *ELF3 Sha-YFP* lines, one slide from each line was prepared, and both slides were put together in the microscope. The plants from each slide were then scanned, one after another, within 30 min and with the same microscope settings. The microscope settings for the *Figure 13* data set were as follows: Image size: x = 512, y = 512, z = 20; Channels: 3, 8-bit, Zoom = 1.0; Objective: Plan-Aprochromat 40x/1.30 Oil; Pixel dwell: 2.55 μs; Master gain: ch1 = 972, ch2 = 847, ch3 = 162; Digital gain: ch1 = 1.20, ch2 = 1.0, ch3 = 1.50; Digital offset = ch1 = −18.0, ch2 = 2.0, ch3 = −24.42; Pinhole = 156 μm; and laser: 488 nm with 10.0% strength. For the *Figure 12* data set, microscope settings were as follows: Image size: x = 512, y = 512, z = 20; Channels: 3, 8-bit, Zoom = 0.5; Objective: Plan-Aprochromat 63x/1.40 Oil; Pixel dwell: 2.55 μs; Master gain: ch1 = 1096, ch2 = 928, ch3 = 393; Digital gain: ch1 = 1.20, ch2 = 0.61, ch3 = 1.40; Digital offset = 0.0, Pinhole = 156 μm; and laser: 488 nm with 10.0% strength.

## Population genetic analysis

The *ELF3* CDS sequences of accessions were downloaded from the 1001 Genome-Project (http://www.1001genomes.org/). The geographical coordinates for the accessions were obtained

from the SALK database (http://signal.salk.edu/atg1001/index.php). These coordinates were used to map the geographical position of the accessions using Google maps (maps.google.com) (*Table 4*). For sequence alignment and phylogenetic analysis, MEGA 4.0 software was used (*Tamura et al., 2007*). The phylogenetic relationships between the sequences were determined using the neighbor-joining (NJ) method and applying the interior–branch test (*Saitou and Nei, 1987*).

## Acknowledgements

We are grateful to C Darrah for the modified pPZP221 vector. We also thank B Pieper, J de Montaigu, D Staiger, A Hörger, and D Martin for comments on the manuscript. K Schneeberger and M von Korff assisted in statistical analyses.

## Additional information

### Funding

| Funder | Grant reference number | Author |
| --- | --- | --- |
| Deutsche Forschungsgemeinschaft | SPP1530 | Muhammad Usman Anwer, Eleni Boikoglou, Eva Herrero, Marc Hallstein, Amanda Melaragno Davis, Geo Velikkakam James, Ferenc Nagy, Seth Jon Davis |
| Deutsche Forschungsgemeinschaft | DA 1061/4-1 | Muhammad Usman Anwer, Eleni Boikoglou, Eva Herrero, Marc Hallstein, Amanda Melaragno Davis, Geo Velikkakam James, Ferenc Nagy, Seth Jon Davis |
| Deutsche Forschungsgemeinschaft | SFB635 | Muhammad Usman Anwer, Eleni Boikoglou, Eva Herrero, Marc Hallstein, Amanda Melaragno Davis, Geo Velikkakam James, Ferenc Nagy, Seth Jon Davis |

The funders had no role in study design, data collection and interpretation, or the decision to submit the work for publication.

### Author contributions

MUA, EB, Conception and design, Acquisition of data, Analysis and interpretation of data, Drafting or revising the article, Contributed unpublished essential data or reagents; EH, Conception and design, Acquisition of data; MH, Acquisition of data, Contributed unpublished essential data or reagents; AMD, Acquisition of data, Drafting or revising the article; GVJ, Conception and design, Acquisition of data, Analysis and interpretation of data; FN, Conception and design, Analysis and interpretation of data, Drafting or revising the article, Contributed unpublished essential data or reagents; SJD, Conception and design, Analysis and interpretation of data, Drafting or revising the article

## Additional files

### Supplementary files

• Supplementary file 1. RIL periodicity of *CCR2::LUC* in BxS after photic entrainment.

• Supplementary file 2. RIL periodicity of *CCR2::LUC* in BxS after thermal entrainment.

• Supplementary file 3. Detail of primers used.

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
