## [Decision Letter]

Thank you for sending your work entitled “An *ELF3* Variant from Central Asia Defines that Localization is Associated to Its Action in the Circadian Clock” for consideration at *eLife*. Your article has been favorably evaluated by a Senior editor (Detlef Weigel), a Reviewing editor, and 2 reviewers, one of whom, Justin Borevitz, has agreed to reveal his identity.

The Reviewing editor and the reviewers discussed their comments before we reached this decision, and the Reviewing editor has assembled the following comments to help you prepare a revised submission.

This elegant manuscript provides an excellent example of how a natural genetic variant cannot only be reduced to a single nucleotide/amino acid, but that considerable new functional knowledge can emanate from such an approach.

You identify a new variant at the ELF3 locus, previously implicated by others in natural genetic variation of shade avoidance responses and circadian clock behavior. However, the precise variant was not identified, and as main contributor a polyglutamine repeat stretch was suggested in one of the studies. The current work shows, however, that the polygulatmine stretch makes only a very minor contribution to ELF3 functional variation.

The functional studies in this manuscript are an important contribution to the plant circadian field as null ELF3 alleles are arrhythmic, precluding detailed analysis of their precise role in the circadian network. Using the natural hypomorphic allele, you now show that ELF3 is indeed a core clock gene that is required for both proper circadian rhythms in the light, and also for sustained rhythms in darkness. Furthermore, you show that ELF3 is required for appropriate entrainment to light/dark cycles, indicating a dual role for this protein in the core clock mechanism as well as in the input pathway that connects light signals to the circadian clock. Finally, you provide evidence that this allele affects the subcellular localization of the protein, interfering with the formation of nuclear speckles and resulting in increased cytoplasmic accumulation of this protein, presumably through alterations in the ability of this ELF3 variant to interact with other clock components that localize to the nucleus such as GI.

1) The evolution section is the weakest of the paper and should either be largely removed. Alternatively, this should be carried out much more rigorously. For example, one would want to know what the size of the selective sweep is, from natural recombination events flanking the 15 or more Shah class accessions. This would need to be compared to selection signatures at background loci (clock genes for example) in these same geographically isolated strains.

2) In the Discussion, there is no need to speculate about a second site modifier as other QTL could buffer the period effect. You could simply conclude that the 362V (ELF3-Sha), so elegantly proven, “has a regional distribution and this warrants further investigation of its potentially local adaptive effects.”

Other issues to address:

3) The studies by Jimenez-Gomez and Undurraga need to be introduced in the Introduction.

4) Strong purifying selection (if this section is maintained): Why was only the coding sequencing retrieved from the 1001 genomes, why not +-15kb around ELF3? How similar are the 15 accessions in a STRUCTURE analysis? What are the *A. lyrata* and *C. rubella* sequences of ELF3, and how can this information be used to ascertain selection at the ELF3 locus? Which allele is ancestral and which is derived? It may be that Bay is derived and now at high frequency, or that Shah is part of an Asian selective sweep.

5) Figure 7 looks like many more the 59 changes; how many are replacement vs synonymous?

6) What is the Kondara seq? It looks like a near neighbor of the Shah class.

7) The average altitude is meaningless. Many alleles from this region would be elevated as it is high country. Also, the histogram is much less informative than a box-and-whiskers plot.

8) Figure 8 doesn't show anything significant about the site in the protein as the Tajima's D is largely negative genome wide.

9) “Our data reveals that the ELF3 locus has been a target of natural selection, which is maintaining its current allelic state by removing recently arisen mutations.” Displacing signature of selective sweep? How common a signature? How many other SNPs show similar latitudinal band?

10) Table: “F shows the variation explained by each factor relative to the error variation.” F is not the variation explained, but the mean SS/error SS; it would be good to show % with total including error adding to 100.

11) “TRANS(Genotype)” What is trans? No need to separate this.

12) LOD score of 2.4 is low, set by permutations of 71 RILs? Genome wide p < 0.01? The largest QTL is ELF3 and well proven but are other QTL real and truly different between entrainment environments?

13) Figure 8 window size is too small, should include +-30kb, and genomic frequency should be reported.

14) Figure 1) color red or blue parental genotype arrows, b) should be a scatter plot TMP vs LD.

---

## [Author Response]

*1) The evolution section is the weakest of the paper and should either be largely removed. Alternatively, this should be carried out much more rigorously. For example, one would want to know what the size of the selective sweep is, from natural recombination events flanking the 15 or more Shah class accessions. This would need to be compared to selection signatures at background loci (clock genes for example) in these same geographically isolated strains*.

As evolution is complex and is often hard to prove, we understand the reviewers concern about the evolution section of the manuscript and agree that this section had been the weakest in the paper. Therefore, in line with the reviewers’ suggestion, we have removed much of this data. Specifically, the entire analyses of selection [Tajima’s D (Figure 8), Fu and Li’ D* and Fu and Li’s F* (Figure 9)] along with the data about the possible association of ELF3-Sha distribution with the altitude (Figure7C) were removed.

*2) In the Discussion, there is no need to speculate about a second site modifier as other QTL could buffer the period effect. You could simply conclude that the 362V (ELF3-Sha), so elegantly proven, “has a regional distribution and this warrants further investigation of its potentially local adaptive effects*.*”*

The text about the second site modifier was deleted and the Discussion was accordingly modified.

Other issues to address:

*3) The studies by Jimenez-Gomez and Undurraga need to be introduced in the Introduction already*.

Both these studies are now cited in the Introduction section.

*4) Strong purifying selection (if this section is maintained): Why was only the coding sequencing retrieved from the 1001 genomes, why not +-15kb around ELF3? How similar are the 15 accessions in a STRUCTURE analysis? What are the* A. lyrata *and* C. rubella *sequences of ELF3, and how can this information be used to ascertain selection at the ELF3 locus? Which allele is ancestral and which is derived? It may be that Bay is derived and now at high frequency, or that Shah is part of an Asian selective sweep*.

The selection section was removed, as requested above. Please see our response to comment 1.

*5)*
Figure 7
*looks like many more the 59 changes; how many are replacement vs synonymous?*

Our careful re-evaluation of the sequence data confirmed that there are indeed 59 polymorphic sites, out of which 43 are replacements and 16 are synonymous. We have also added this text in the Results: “Out of these 59 polymorphic sites, 16 were synonymous and 43 were nonsynonymous”.

The details are as follows: Segregating sites: 59; Total number of Synonymous changes: 16

Position in CDS: 249 423 519 522 717 813 873 918 1104 1269 1476 1536 1773 1809 1977 2034

Total number of Replacement changes: 43

Position in CDS: 21 122 199 202 211 256 302 343 416 461 470 510 515 574 580 664 700 701 715 736 805 849 865 883 922 964 1033 1034 **1085 (ELF3-Sha)** 1204 1288 1512 1551 1558 1628 1691 1700 1732 1772 1874 1895 1907 1982

Please note that we have updated the figure without noteworthy changes in the Results interpretation. The updated phylogenetic tree is now based on the nucleotide sequence of *ELF3* as oppose to the amino-acid sequence used in the previous figure (Please see Figure 14).

*6) What is the Kondara seq? It looks like a near neighbor of the Shah class*.

Kondara *ELF3* Sequence is same as that of Sha, except for a single base-pair deletion in Kondara at position 1628. In the updated phylogenetic analysis (Figure 14), the sites with gaps/missing data were not included in the calculations. Please see our response to comment 5.

*7) The average altitude is meaningless. Many alleles from this region would be elevated as it is high country. Also, the histogram is much less informative than a box-and-whiskers plot*.

We agree with the reviewers on this point. All data related to average altitude is removed.

*8)*
Figure 8
*doesn't show anything significant about the site in the protein as the Tajima's D is largely negative genome wide*.

Figure 8 along with Figure 9 are now deleted. This was in response to the request to delete this section.

9) “Our data reveals that the ELF3 locus has been a target of natural selection, which is maintaining its current allelic state by removing recently arisen mutations.” Displacing signature of selective sweep? How common a signature? How many other SNPs show similar latitudinal band?

All data related to selection is removed. Please see our response to comment 1.

*10) Table: “F shows the variation explained by each factor relative to the error variation.” F is not the variation explained, but the mean SS/error SS; it would be good to show % with total including error adding to 100*.

The definition of “F” was modified as suggested. We could not fully understand what the reviewers meant by “it would be good to show % with total including error adding to 100.” Because of this, we have not yet included this information. If the editor requires this change, we believe we can easily provide this information if the reviewer can more fully explain what is requested**.**

*11) “TRANS(Genotype)” What is trans? No need to separate this*.

Trans stands for ‘Transformants’ within each genotype (RIL). This information is also deleted as suggested (Table 2).

12) LOD score of 2.4 is low, set by permutations of 71 RILs? Genome wide p < 0.01? The largest QTL is ELF3 and well proven but are other QTL real and truly different between entrainment environments?

The LOD threshold score was calculated by thrice performing the permutation test using 1% genome-wide significance in MapQTL5.0. After each permutation test, different LOD values along with their significance level were obtained by MapQTL5.0. Only the LOD value corresponding to a p<0.01 was selected in each permutation test. The three LOD values obtained by each permutation test were then averaged to gain an overall genome-wide LOD of 2.4. As the LOD scores obtained by permutation test are based on the original phenotypic data (here circadian periodicity), it can vary depending upon the data quality and trait under study. As such, the LOD2.4 obtained in this study is not significantly low.

Similar LOD threshold (2.55) has been reported by Darrah et al. Plant Phys., for circadian phase using 65 RILs. In many other studies, similar LOD thresholds have also been reported for other traits such as flowering time (El-Lithy et al. Genetics 2006, LOD 2.4) and seed size (Herridge et al. Plant Methods 2011, LOD 2.35), as others' reported examples. Further having the LOD threshold at 2.4 does not affect the result of the analysis as all of the QTL detected scored much higher than 2.4. The lowest LOD score of 3.21 was obtained for Chr1 QTL (Table 3), which is still considerably higher than the threshold 2.4.

As the Chr2 QTL displayed the largest effect, we pursued its validation. This was taken to the level of QTL identification. The molecular identity of the other QTLs can be confirmed in subsequent studies.

*13)*
Figure 8
*window size is too small, should include +-30kb, and genomic frequency should be reported.*

This figure is now removed. Please see our response to comment 1.

*14)*
Figure 1*) color red or blue parental genotype arrows, b) should be a scatter plot TMP vs LD.*

a) The parental genotypes arrows are now colored in Red, b) we have added a scatter plot for TMP vs LD periodicities (Figure 1). We have also kept the graph now Figure 1, as it provides additional information about the LD Vs TMP periodicity difference of the individual RILs.